EMBO
Molecular Medicine

# Reversible p53 inhibition prevents cisplatin ototoxicity without blocking chemotherapeutic efficacy

Nesrine Benkafadar[1,2,†], Julien Menardo[1,2,†], Jérôme Bourien[1,2], Régis Nouvian[1,2], Florence François[1,2], Didier Decaudin[3], Domenico Maiorano[4], Jean-Luc Puel[1,2] & Jing Wang[1,2,*]

## Abstract

Cisplatin is a widely used chemotherapy drug, despite its significant ototoxic side effects. To date, the mechanism of cisplatin-induced ototoxicity remains unclear, and hearing preservation during cisplatin-based chemotherapy in patients is lacking. We found activation of the ATM-Chk2-p53 pathway to be a major determinant of cisplatin ototoxicity. However, prevention of cisplatin-induced ototoxicity is hampered by opposite effects of ATM activation upon sensory hair cells: promoting both outer hair cell death and inner hair cell survival. Encouragingly, however, genetic or pharmacological ablation of p53 substantially attenuated cochlear cell apoptosis, thus preserving hearing. Importantly, systemic administration of a p53 inhibitor in mice bearing patient-derived triple-negative breast cancer protected auditory function, without compromising the anti-tumor efficacy of cisplatin. Altogether, these findings highlight a novel and effective strategy for hearing protection in cisplatin-based chemotherapy.

**Keywords** chemotherapy; cisplatin ototoxicity; DNA damage; hearing loss; protection

**Subject Categories** Cancer; Neuroscience

## Introduction

Cisplatin (*cis*-diammine dichloroplatinum(II); CDDP) is a highly effective and widely used chemotherapeutic agent for the treatment of different types of human tumors, particularly solid tumors (Siddik, 2003; Wang & Lippard, 2005). Unfortunately, CDDP has a number of dose limiting side effects including ototoxicity, which greatly hamper its chemotherapeutic efficacy (Fram, 1992; Hartmann & Lipp, 2003). Indeed, CDDP has been shown to induce degeneration of the neuro-sensory epithelium of the cochlea with partial or complete loss of sensory outer hair cells (OHCs) and a sporadic loss of inner hair cells (IHCs) and spiral ganglion neurons (SGNs), resulting in irreversible hearing loss (Anniko & Sobin, 1986; Tsukasaki *et al*, 2000; Wang *et al*, 2003a, 2004). The mechanism of CDDP-induced ototoxicity remains unclear, yet its understanding could lead to measures to protect hearing in patients undergoing CDDP-based chemotherapy.

In tumors and cancer cells, CDDP-induced DNA damage has been recognized as the major cause of cell injury and death during chemotherapy (Rabik & Dolan, 2007). CDDP forms covalent bonds with the purine bases in DNA, primarily resulting in intra-strand cross-linking, which blocks gene transcription and potentially results in double-strand breaks (Jung & Lippard, 2003; Wang & Lippard, 2005). In response to DNA damage, three major molecular sensors called ATM (ataxia telangiectasia mutated), ATR (ataxia telangiectasia and Rad3-related), and DNA-PK (DNA-dependent protein kinase) are recruited to the site of DNA damage, forming nuclear foci (Ciccia & Elledge, 2010; Smith *et al*, 2010). These sensors then phosphorylate several proteins to induce the DNA damage response (Bassing *et al*, 2002; Celeste *et al*, 2002), in which p53, a major tumor suppressor, plays a pivotal role. Downstream of this response are targets which either promote cell survival or cause cells to undergo apoptosis (Shiloh, 2006; Di Micco *et al*, 2008).

In the cochlea, it remains unclear whether the same DNA damage response pathways are triggered by CDDP and how they are involved in cochlear cell apoptosis. Consequently, it remains to be determined whether it is possible to block pathways responsible for cochlea toxicity without diminishing the chemotherapeutic effect of CDDP.

Here, we deciphered the CDDP-toxicity pathway in the organ of Corti at both cellular and systemic levels and revealed the

1 INSERM - UMR 1051, Institut des Neurosciences de Montpellier, Montpellier, France
2 Université de Montpellier, Montpellier, France
3 Laboratoire d'Investigation Pré -Clinique/Service d'Hématologie Clinique, Institut Curie, Paris, France
4 CNRS - UPR1142, Institut de Génétique Humaine, Montpellier, France
*Corresponding author. Tel: +33 499 636 048; Fax: +33 499 636 020; E-mail: jing.wang@inserm.fr
†These authors contributed equally to this work

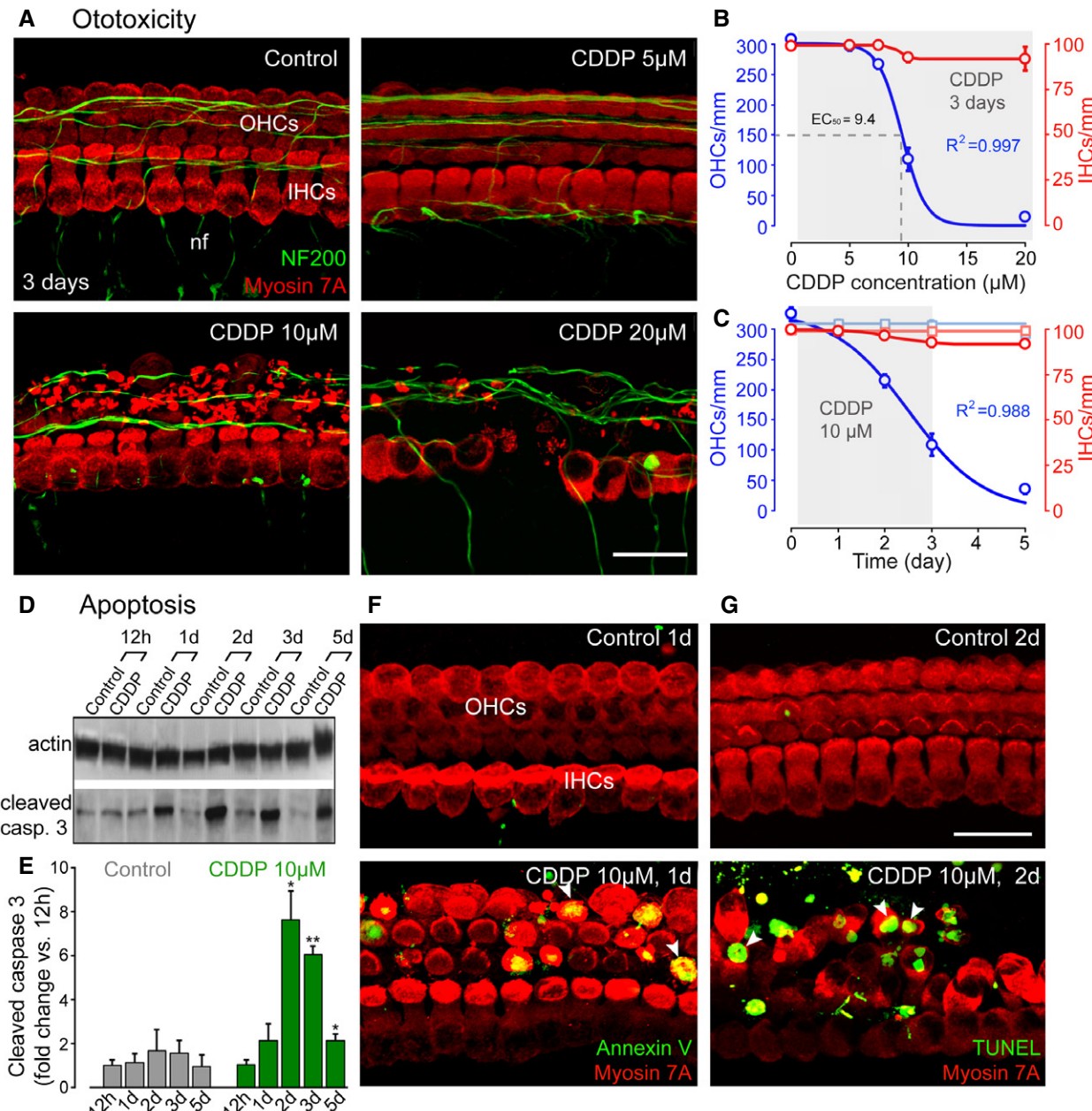

**Figure 1.  Dose- and time-dependent CDDP-induced apoptosis of sensory hair cells in organ of Corti cultures.**

A   Confocal images showing the basal regions of cochlear explants treated with either culture medium alone or medium containing 5, 10 or 20 μM CDDP for 3 days. The explants were labeled with myosin 7A (red) to identify hair cells and NF200 (green) to highlight auditory nerve fibers. Scale bar = 20 μm. nf: nerve fibers, OHCs: outer hair cells, IHCs: inner hair cells.

B   Dose–response curves of CDDP-induced loss of OHCs (blue line) and IHCs (red line) in basal cochlear regions. Data are expressed as mean ± SEM ($n$ = 5 cochleae per condition and per time point).

C   Effect over time on OHCs and IHCs treated with either culture medium alone (light blue and red lines for OHCs and IHCs, respectively) or 10 μM CDDP (blue and red lines for OHCs and IHCs, respectively). Data are expressed as mean ± SEM ($n$ = 5 cochleae per condition and per time point).

D   Representative Western blot analysis using antibodies against β-actin and cleaved caspase-3 in whole cochlear extracts.

E   Histogram representing the change in cleaved caspase-3 expression levels over time in control and CDDP groups ($n$ = 6 cochleae per condition and per time point). Actin served as a loading control. Data are expressed as mean ± SEM. One-way ANOVA test followed by *post hoc* Tukey's test (*$P \leq 0.028$, **$P = 0.0007$, cochleae at the different times after CDDP exposure versus control 12 h).

F, G   Confocal images of cochlear explants treated with either culture medium alone or containing 10 μM CDDP for 1 (F) or 2 (G) days. Hair cells were identified using myosin 7A (red), phosphatidylserine sites on the cell membrane surface were detected using fluorochrome-labeled Annexin V (green in F), and apoptotic DNA fragmentation was identified using a TUNEL apoptosis kit (green in G). The white arrowheads indicate cell surface Annexin V-positive labeling (lower left) and TUNEL-positive nuclei (lower right). Scale bar = 15 μm.

Data information: All experiments were performed in triplicate. In (B) and (C), continuous lines are sigmoidal fits to the data.

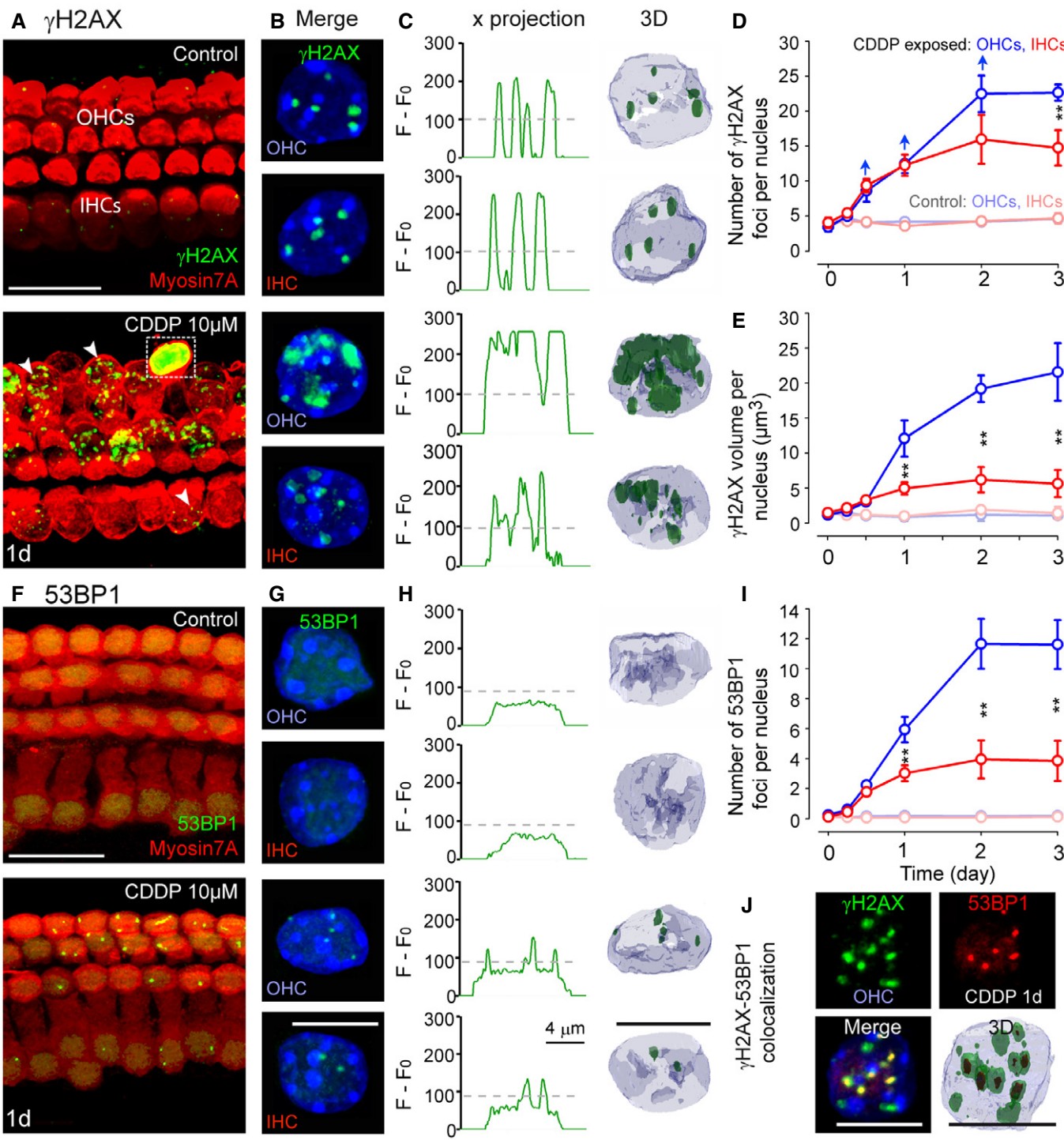

**Figure 2.**

ATM-Chk2-p53 pathway as the major CDDP-activated cascade. Targeting this p53 signaling pathway attenuated cochlear hair cell death and preserved hearing function during CDDP treatment, without interfering with the chemotherapeutic efficacy. Altogether, this highlights pharmacological inhibition of p53 as an attractive approach toward improving the safety of CDDP-based chemotherapy.

# Results

## CDDP ototoxicity is predominantly the result of OHC apoptosis

Previous reports have shown that CDDP elicits a substantial loss of OHCs in the basal turn of the cochlea, with a lower impact on IHCs and no obvious effect on apical hair cells (Langer *et al*, 2013).

    

**Figure 2. Distribution of γH2AX and 53BP1 foci induced by CDDP in IHCs versus OHCs.**

A    Confocal images showing the basal region of organ of Corti cultures treated with either culture medium alone or medium containing 10 μM CDDP for 1 day and immunolabeled for myosin 7A (red) and γH2AX (green). The white arrowheads indicate CDDP-induced increase in γH2AX foci in both OHCs and IHCs. The white box shows a dying OHC with the condensed cytoplasm and γH2AX pan-stained nucleus. Scale bar = 24 μm.

B    Higher magnification images of representative OHC and IHC nuclei from all conditions tested in (A). Scale bar = 5 μm.

C    Histograms displaying green fluorescent signal intensity of x-projections and 3D reconstruction images from OHC and IHC nuclei presented in (B). F0 corresponds to background noise, and gray dashed lines represent the threshold used to detect specific foci labeling. 3D images were reconstructed according to the threshold defined in the histograms of all conditions tested. Scale bar = 5 μm.

D, E    Quantification analysis of γH2AX foci number (D) and total volume of foci per nucleus (E) from OHCs and IHCs treated with either culture medium alone (light blue and red lines for OHCs and IHCs, respectively) or 10 μM CDDP (blue and red lines for OHCs and IHCs, respectively; n = 50 nuclei per condition and per time point). The blue arrows (in D) indicate the possibility of underestimation of counted foci in OHCs using three-dimensional (3D) reconstruction image analysis. Data are expressed as mean ± SEM. One-way ANOVA test followed by post hoc Tukey's test (D: **P = 0.007; E: **P ≤ 0.009; CDDP-exposed OHCs versus CDDP-exposed IHCs).

F    Confocal images showing the basal region of organ of Corti cultures treated with either culture medium alone or medium containing 10 μM CDDP for 1 day and immunolabeled for myosin 7A (red) and 53BP1 (green). Scale bar = 24 μm.

G    Higher magnification images of representative OHC and IHC nuclei from all conditions tested in (F). Scale bar = 5 μm.

H    Histograms displaying green fluorescent signal intensity of x-projections and 3D reconstruction images from OHC and IHC nuclei presented in (G). F0 corresponds to background noise, and gray dashed lines represent the threshold used to detect specific foci labeling. 3D images were reconstructed according to the threshold defined in the histograms of all conditions tested. Scale bar = 5 μm.

I    Quantification analysis of 53BP1 foci number from OHCs and IHCs treated with either culture medium alone (light blue and red lines for OHCs and IHCs, respectively) or 10 μM CDDP (blue and red lines for OHCs and IHCs, respectively; n = 50 nuclei per condition and per time point). Data are expressed as mean ± SEM. One-way ANOVA test followed by post hoc Tukey's test (**P ≤ 0.009; CDDP-exposed OHCs versus CDDP-exposed IHCs).

J    Higher magnification images showing representative OHC nuclei from the basal region of the organ of Corti treated with 10 μM CDDP for 1 day and immunolabeled for 53BP1 (red), γH2AX (green), and counterstained with Hoechst (blue). The 3D image shows co-localization of smaller sized 53BP1 foci within the γH2AX foci. Scale bars = 5 μm.

Data information: All experiments were performed in triplicate.

Concordantly, using explants of organ of Corti (Fig EV1A), we consistently found that 3 days exposure to micromolar concentrations of CDDP led to a massive and concentration-dependent degeneration of OHCs in the basal turn of the organ of Corti ($EC_{50}$ = 9.4 μM, Fig 1A and B). We also observed only a marginal loss of IHCs (Fig 1A and B). Remaining hair cell counts after exposure to a CDDP concentration close to the $EC_{50}$ (10 μM) showed a time-dependent OHC loss with a drastic disappearance 2 days following CDDP treatment (day 5). By contrast, the IHCs were minimally affected (Fig 1C). In addition, we observed no significant decrease in spiral ganglion neuron density in the cochlear slices 2 days following CDDP treatment (day 5) (Fig EV1C and D). Therefore, all the quantified data presented in our study are focused on the hair cells of the basal cochlear turn.

To determine the nature of this CDDP-induced OHC loss, we firstly probed for apoptotic markers. Western blots from protein extracts of CDDP-exposed cochleae revealed high levels of caspase-3 activation (Fig 1D and E). Concomitantly, CDDP-exposed organs of Corti displayed a predominance of cell surface annexin V-positive labeling (Fig 1F) and TUNEL-positive nuclei (Fig 1G) in the OHCs from the basal cochlear turn of the cochlea, when compared with controls (Fig 1F and G). In contrast, only marginal IHCs showed positive annexin V or TUNEL staining (Fig EV1F and G). Altogether, these results demonstrate that apoptotic pathways are responsible for the CDDP-induced hair cell loss.

### γH2AX and 53BP1 signal DNA damage induced by CDDP

To investigate the role of CDDP-induced DNA damage in this apoptosis, we used immunolabeling to identify within hair cell nuclei two hallmarks of DNA damage: H2AX phosphorylation (γH2AX) and 53BP1. In the organ of Corti control cultures, the majority of IHCs and OHCs displayed similar basal levels of γH2AX foci (Fig 2A and B). CDDP intoxication greatly increased the formation of γH2AX foci in both types of sensory hair cell nuclei, although OHCs tended to exhibit higher numbers of γH2AX foci (Fig 2A and B). The immunolabeled foci observed in CDDP-damaged OHCs were also larger in size (Fig 2A and B) which may have led to difficulties in distinguishing individual foci and possibly underestimated the foci count using three-dimensional (3D) reconstruction image analysis (Fig 2C and D). We therefore estimated the total volume of the γH2AX foci immunolabeling per nucleus (Fig 2E). The total γH2AX foci size increased over the 3 days of CDDP exposure within the nuclei of both hair cell types. The OHCs however displayed a significantly larger γH2AX foci volume in comparison with IHCs (Fig 2E).

53BP1 is a central regulator of double-strand break signaling, allowing the tracking of double-strand break responses (Rogakou et al, 1998; Schultz et al, 2000). While none of the control cultures displayed any 53BP1 foci, CDDP-intoxicated cultures showed a progressive increase over the exposure period in the nuclei of both hair cell types (Fig 2F–I). The OHC nuclei displayed a higher number of 53BP1 spots than the IHC nuclei (Fig 2I). Finally, to determine whether both γH2AX and 53BP1 are recruited to the same DNA damage sites, we performed double-staining experiments (Fig 2J). Although the majority of γH2AX foci did not co-localize with 53BP1 foci, a substantial fraction were closely associated, most likely reflecting double-strand breaks (21.2 ± 2.4% and 34.7 ± 5.5% in IHC and OHC nuclei, respectively). In contrast, only some spiral ganglion neurons from the CDDP-intoxicated cochlear slices displayed a low level of nuclear γH2AX and 53BP1 foci formation (Fig EV1H–I).

### ATM-dependent signaling pathway leads to hair cell loss

Double-strand breaks can lead to ATM activation and subsequent Chk2 phosphorylation (Smith et al, 2010) and/or ATR activation,

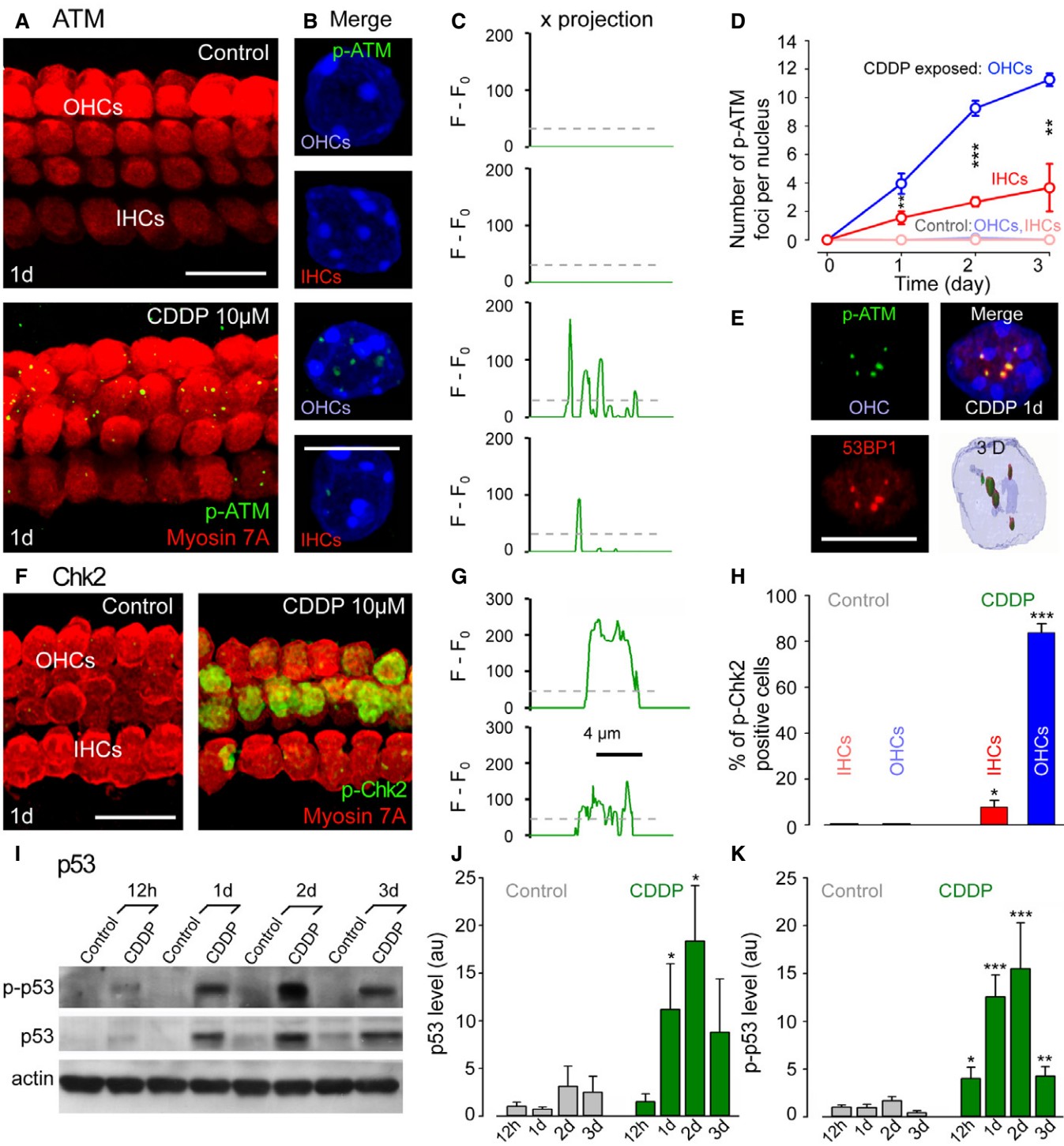

**Figure 3.**

resulting in phosphorylation of Chk1 and p53 (Marazita *et al*, 2012). Although we found both ATR and Chk1 mRNA present in the cochlea, we failed to detect Chk1 phosphorylation (p-Chk1) during CDDP intoxication in cochlear cultures (Fig EV2A and B). Therefore, we investigated whether double-strand breaks induced by CDDP result in activation of the ATM pathway. In contrast to the organ of Corti control cultures, which showed negative staining for phosphorylated forms of ATM (p-ATM) in both OHCs and IHCs (Figs 3A–C and EV2C), CDDP intoxication greatly increased the number of p-ATM foci in both types of hair cell nuclei (Figs 3A–C and EV2C). In concordance with the higher numbers of γH2AX and 53BPI foci in OHC nuclei, we also found higher levels of p-ATM foci in OHC nuclei compared to IHC nuclei (Fig 3D). Finally, co-immunolabeling experiments (Fig 3E)

**Figure 3.  Activation of the ATM-Chk2-p53 pathway upon CDDP treatment.**

A    Confocal images showing the basal region of organ of Corti cultures treated with either culture medium alone or medium containing 10 μM CDDP for 1 day and immunolabeled for myosin 7A (red) and p-ATM (green). Scale bar = 15 μm.

B    Higher magnification images showing representative OHC and IHC nuclei from control and CDDP-exposed cells. Scale bar = 5 μm.

C    Histograms displaying green fluorescent signal intensity of x-projections of OHC and IHC nuclei from control and CDDP-exposed cells.

D    Quantification analysis of p-ATM foci numbers in both OHCs and IHCs exposed to either culture medium alone (light blue and red lines for OHCs and IHCs, respectively) or 10 μM CDDP (blue and red lines for OHCs and IHCs, respectively; n = 50 nuclei per condition and per time point). Data are expressed as mean ± SEM. One-way ANOVA test followed by post hoc Tukey's test (**P < 0.008, ***P = 0.00005; CDDP-exposed OHCs versus CDDP-exposed IHCs).

E    Higher magnification confocal and 3D images showing the co-localization of p-ATM foci (green) with 53BP1 foci (red) in an OHC nucleus from the basal region of the organ of Corti culture treated with 10 μM CDDP for 1 day. Scale bar = 5 μm.

F    Confocal images showing the basal region of organ of Corti cultures treated with either culture medium alone or medium containing 10 μM CDDP for 1 day and immunolabeled for myosin 7A (red) and p-Chk2 (green). Scale bar = 15 μm.

G    Histograms of green fluorescent signal intensity from OHC and IHC nuclei of a CDDP-treated organ of Corti explant.

H    Quantification of Chk2-positive nuclei in both IHCs (red bars) and OHCs (blue bars) from control and CDDP-treated organ of Corti cultures (n = 50 nuclei per condition and per time point). Data are expressed as mean ± SEM. Kruskal–Wallis test followed by post hoc Dunn's test (*P = 0.045, ***P = 0.0002; CDDP-exposed IHCs versus control IHCs, CDDP-exposed OHCs versus control OHCs).

I    Representative Western blots using antibodies against p-p53 (serine 15), p53, and β-actin in control and 10 μM CDDP-exposed whole cochlear extracts.

J, K  Histograms representing the levels of total p53 protein (J) and p53 phosphorylation (K) in control (gray bars) and CDDP-exposed cochleae (green bars; n = 6 cochleae per condition and per time point). Actin served as a loading control. Data are expressed as mean ± SEM. One-way ANOVA test followed by post hoc Tukey's test (J: *P ≤ 0.023; K: *P = 0.032, **P = 0.007, ***P ≤ 0.0008; cochleae at the different times after CDDP exposure versus control 12 h).

Data information: All experiments were performed in triplicate.

demonstrated a clear overlap between p-ATM and 53BP1 foci ($96.1 \pm 2.7\%$ and $92.6 \pm 1.8\%$ of co-localizations in IHC and OHC nuclei, respectively), suggesting that the ATM activation occurs at double-strand break sites.

We subsequently analyzed the activation of two key ATM targets: Chk2 and p53. One day post-CDDP intoxication, we found p-Chk2 pan-staining primarily in OHC nuclei ($83.9 \pm 3.8\%$ positive nuclei), and in a smaller percentage of IHC nuclei ($7.7 \pm 3.1\%$ positive nuclei; Figs 3F–H and EV2D). Next, we studied the activation of p53, as reflected by its accumulation and phosphorylation. CDDP intoxication also greatly increased both the expression and phosphorylation of p53 in the cochlea (Fig 3I–K). However, in our experimental conditions, we failed to detect any activation of ATM or Chk2 in spiral ganglion neurons after CDDP intoxication.

### Opposite effects of ATM activation on IHCs and OHCs

We investigated the role of ATM activation in DNA damage responses and finally hair cell death using a specific ATM inhibitor, KU55933. As expected, KU55933 reduced ATM activation, that is, we found fewer p-ATM foci in both hair cell nuclei during CDDP exposure (Fig EV2E and F). This inhibition of ATM activation was associated with a significant increase in 53BP1 foci in CDDP-exposed IHC nuclei. In contrast, ATM inhibition reduced the number of 53BP1 foci in CDDP-exposed OHC nuclei (Fig 4A and B). Upon analysis of hair cell survival (Fig 4C and D), we observed a slight, though significant, increase in CDDP-induced IHC loss by the addition of KU55933 (Fig 4D), in accordance with a higher number of 53BP1 foci in the IHC nuclei. Conversely, inhibition of ATM activation preserved almost twice as much OHCs compared to CDDP treatment alone (Fig 4C and D). Overall, these results suggest that activation of ATM plays opposite roles in OHCs and IHCs. In IHC nuclei, it appears that activating the ATM pathway favors double-strand break repair, whereas apoptosis is initiated in OHCs, leading to cell death.

### Inhibition of p53 protects both hair cell types against CDDP ototoxicity in vitro

To investigate the inhibition of p53, a substrate located downstream of ATM, we used pifithrin-α (PFT-α), a reversible inhibitor of p53-mediated apoptosis. Pre-treatment with PFT-α had no significant effect on 53BP1 foci formation in either type of hair cell nuclei (Fig 4E and F). In contrast, PFT-α significantly increased the survival of both hair cell types in CDDP-intoxicated organs of Corti (Fig EV2G and H). Therefore, temporary suppression of p53 function by PFT-α could be a promising approach in reducing the ototoxic side effects of CDDP in cancer therapy.

### Genetic and pharmacological deletion of p53 prevents CDDP ototoxicity in adult mice

Using mice genetically deleted for p53, we tested the validity of our in vitro results on a whole in vivo system. We assessed auditory function by recording auditory brainstem responses (ABRs), which reflect the synchronous activation of auditory neurons from the cochlea up to the colliculi in response to incoming sound. In p53 wild-type (wt) mice, CDDP administration significantly increased ABR thresholds at all frequencies tested (Fig 5A and B). Interestingly, p53 knockout (−/−) mice were clearly more resistant to CDDP intoxication. The average ABR thresholds from 4 to 32 kHz were 66.8 dB SPL ± 2.7 versus 48.7 dB SPL ± 2.8 in CDDP-treated p53wt and p53$^{-/-}$ mice, respectively (Fig 5C). In p53wt mice, CDDP injection provoked a massive OHC loss along a basal-apical gradient, accounting for the ABR threshold increase. Consistent with a better preservation of auditory threshold, higher OHC survival was observed in the basal region of CDDP-treated p53$^{-/-}$ mice compared to p53wt (Fig 5D and E) with only very little IHC loss restricted to the basal cochlear turn. OHC survival rates from p53wt and p53$^{-/-}$ mice were $24.7\% \pm 3.6$ and $67.5\% \pm 0.9$, respectively, at a frequency of 32 kHz.

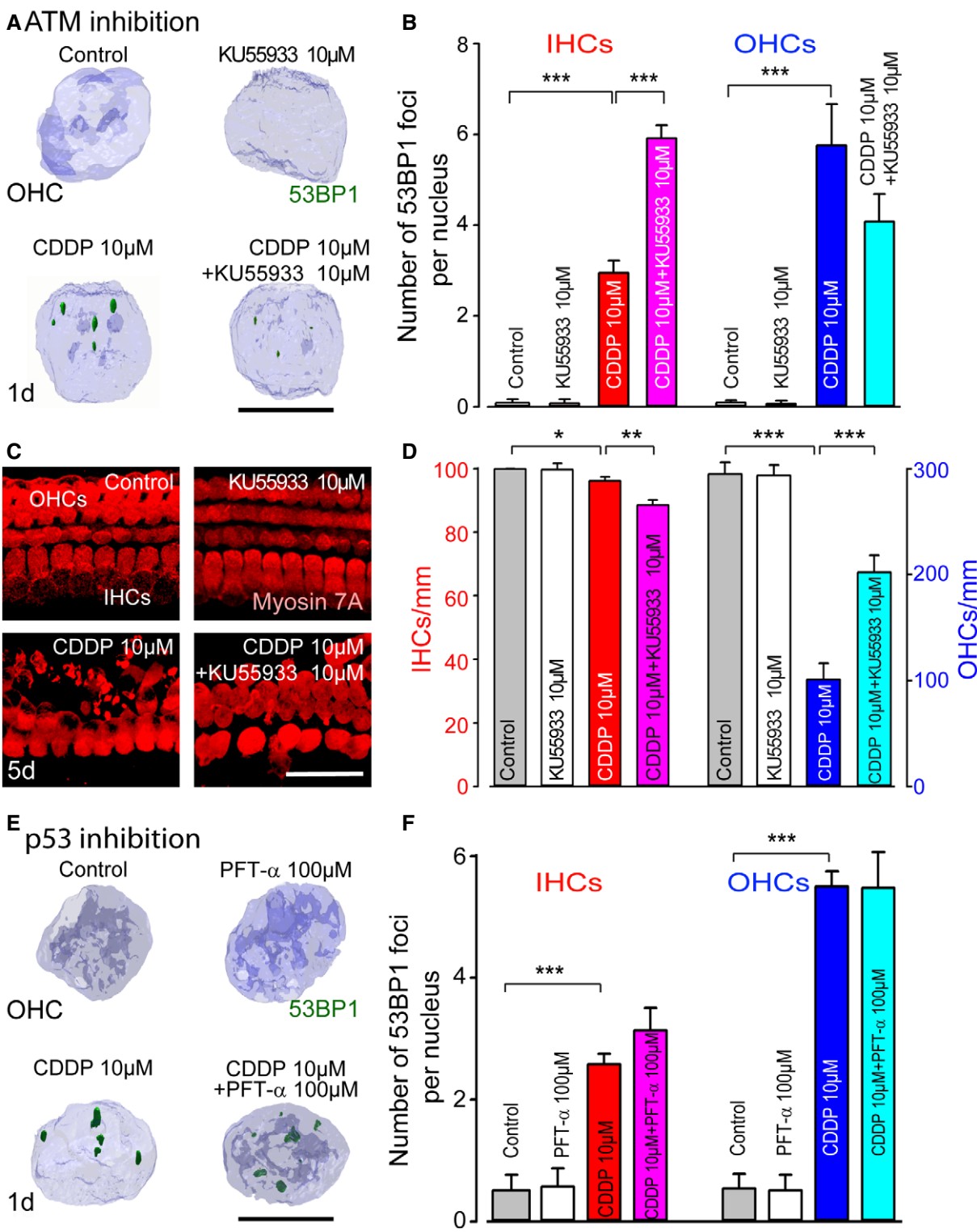

**Figure 4.**

To test the potential usefulness of pharmacological therapies *in vivo*, we used the reversible p53 inhibitor, PFT-α. We showed that a systemic injection of PFT-α prevented an increase in the ABR threshold caused by CDDP administration (mean threshold: 76 dB ± 2.5 versus 49.9 dB ± 2.3 in CDDP + DMSO- and CDDP + PFT-α-treated mice, respectively; Fig 5F and G). Accordingly, morphological examination revealed greater OHC survival in the basal region of CDDP + PFT-α-treated p53wt mice than CDDP + DMSO-treated p53wt mice (OHC survival rates at a frequency of 32 kHz were 71.8% ± 0.65 and 17% ± 1.8, respectively; Fig 5H).

**Figure 4.  Effect of ATM or p53 inhibition on DNA damage repair and hair cell survival.**

A   3D images of OHC nuclei taken from the basal regions of organ of Corti cultures treated with medium alone, 10 μM KU55933, 10 μM CDDP, or 10 μM CDDP in combination with 10 μM KU55933 for 1 day and immunolabeled for 53BP1 (green) and Hoechst 33342 (blue). Scale bar = 5 μm.

B   Quantification analysis of 53BP1 foci number per nucleus in both IHCs (red bars) and OHCs (blue bars) for all conditions ($n$ = 50 nuclei per condition and per time point). Data are expressed as mean ± SEM. One-way ANOVA followed by *post hoc* Tukey's test (***$P$ ≤ 0.0004; CDDP versus control, or CDDP versus CDDP + KU55933).

C   Confocal images showing the basal region of the organ of Corti cultures treated with medium alone, 10 μM KU55933, 10 μM CDDP, or 10 μM CDDP in combination with 10 μM KU55933 for 5 days and immunolabeled for myosin 7A (red). Scale bar = 24 μm.

D   Histograms representing the numbers of surviving IHCs (red bars) and OHCs (blue bars) for all treatment conditions after 5 days ($n$ = 5 cochleae per condition and per time point). Data are expressed as mean ± SEM. One-way ANOVA test followed by *post hoc* Tukey's test (*$P$ = 0.03, **$P$ = 0.004, ***$P$ ≤ 0.0008; CDDP versus control, or CDDP versus CDDP + KU55933).

E   3D images of OHC nuclei from the basal regions of organ of Corti cultures treated with culture medium alone, 100 μM PFT-α, 10 μM CDDP, or 10 μM CDDP in combination with 100 μM PFT-α for 1 day and immunolabeled for 53BP1 (green) and Hoechst 33342 (blue). Scale bar = 5 μm.

F   Quantification of 53BP1 foci number per nucleus in both IHCs (red bars) and OHCs (blue bars) for all conditions ($n$ = 50 nuclei per condition and per time point). Data are expressed as mean ± SEM. One-way ANOVA test followed by *post hoc* Tukey's test (***$P$ ≤ 0.0007; CDDP versus control).

Data information: All experiments were performed in triplicate.

We next wished to identify a protective strategy that would be suitable for clinical use, that is, without impeding the anticancer efficacy of systemically administered CDDP. To this end, we evaluated the efficiency of a local application of PFT-α, through an intratympanic injection using microendoscopy (Fig EV3A and B) also see (Bozzato *et al*, 2014). Our results showed a significant hearing protection in PFT-α-injected right ears (mean threshold: 47.6 dB ± 1.47) when compared to DMSO-injected left ears (68.3 dB ± 1.89) from the same CDDP-intoxicated mice (Fig 5I and J). Consistently, we observed a greater OHC survival in the basal region of PFT-α-injected right ears (OHC survival rates at a frequency of 32 kHz: 77% ± 2.1) than in that of DMSO-injected left ears (20% ± 4.1, Fig 5K).

**Pharmacological inhibition of p53 in triple-negative human breast cancer xenograft mice**

To further extend the versatility of the PFT-α action, we checked whether its systemic administration would interfere with the anticancer effects of CDDP. For this purpose, we used two xenograft models of triple-negative human breast cancer with either wt (HBCx-90) or mutant (HBCx-14) p53 status.

**Inhibition of p53 prevents CDDP ototoxicity without compromising anti-tumor efficacy of CDDP in *TP53* wt tumor-bearing mice**

Firstly, we monitored hearing function in the *TP53* wt HBCx-90 tumor-bearing mice (HBCx-90). As expected, mice receiving either DMSO or PFT-α alone developed neither hearing loss nor hair cell damage (Fig 6A–C). In contrast, the CDDP-treated mice showed a significant increase in ABR thresholds at all frequencies tested (mean threshold: 72.9 dB SPL ± 2.9) in addition to hair cell loss (29.3% ± 3.6 OHC survival at the frequency of 25 kHz, Fig 6A–C). However, systemic injection of CDDP plus PFT-α preserved both auditory function (mean threshold: 49 dB SPL ± 2) and hair cell survival (85% ± 5.8 OHC survival at the frequency of 25 kHz, Fig 6A–C).

An important endpoint in the evaluation of *in vivo* anti-tumor efficacy is the tumor growth inhibitory effect over a long period of time. We observed a partial tumor growth inhibitory effect of CDDP in *TP53* wt HBCx-90-bearing mice (Fig 6D). This is consistent with previous reports showing that human breast cancers with *TP53* wt

are more resistant to doxorubicin or a combination of epirubicin and cyclophosphamide regimen (Bertheau *et al*, 2007; Jackson *et al*, 2012). Importantly, however, the tumor growth rates observed in this tumor model were similar regardless of treatment arm (i.e. CDDP alone or in combination with PFT-α, Fig 6D).

**Inhibition of p53 prevents CDDP ototoxicity, while enhancing anti-tumor efficacy of CDDP in *TP53*-mutant tumor-bearing mice**

We observed a better preservation of auditory threshold and OHC survival in CDDP and PFT-α-treated *TP53*-mutant HBCx-14 tumor-bearing mice (mean threshold: 48.4 dB SPL ± 5.5, and 80.1% ± 4.8 OHC survival at the frequency of 25 kHz), when compared with CDDP alone (mean threshold: 77.7 dB SPL ± 4.3, and 20.8% ± 4.5 OHC survival at the frequency of 25 kHz, Fig 6E–G).

In *TP53*-mutant HBCx-14-bearing mice, while the DMSO or PFT-α alone treatment groups showed continuous tumor growth over a period of 5 weeks, the CDDP-treated group showed significant tumor shrinkage (Fig 6H). Surprisingly, the combination of CDDP with PFT-α demonstrated a greater response to treatment as shown by the complete disappearance of tumors in the majority of mice (8 out of 10 treated mice) and the large decrease in tumor volume in the two remaining mice at day 35 (Fig 6H). In addition, following the last dose of combined CDDP plus PFT-α, tumor growth arrest in these mice was sustained for up to 8 weeks (day 70). This arrest period is much longer than that observed in the CDDP alone group in which tumor regrowth began just 3–4 weeks after the end of treatment (d35-d42, Fig 6H).

Consistently, the histological examination of the tumors collected 1 week after the end of treatment (d21, Fig EV3C and D) revealed that the combined treatment of CDDP plus PFT-α was more efficient in replacing tumor cells by fibrous scar (Fig EV3C and D).

**Reversible p53 inhibition enhanced CDDP-induced anti-angiogenesis and apoptosis selectively in the *TP53*-mutant HBCx-14 tumors**

One mechanism which may account for the heightened *TP53*-mutant tumor sensitization to CDDP/PFT-α treatment could be an alteration in tumor angiogenesis. To test this hypothesis, tumor angiogenesis was evaluated in the most vascular part of CD31

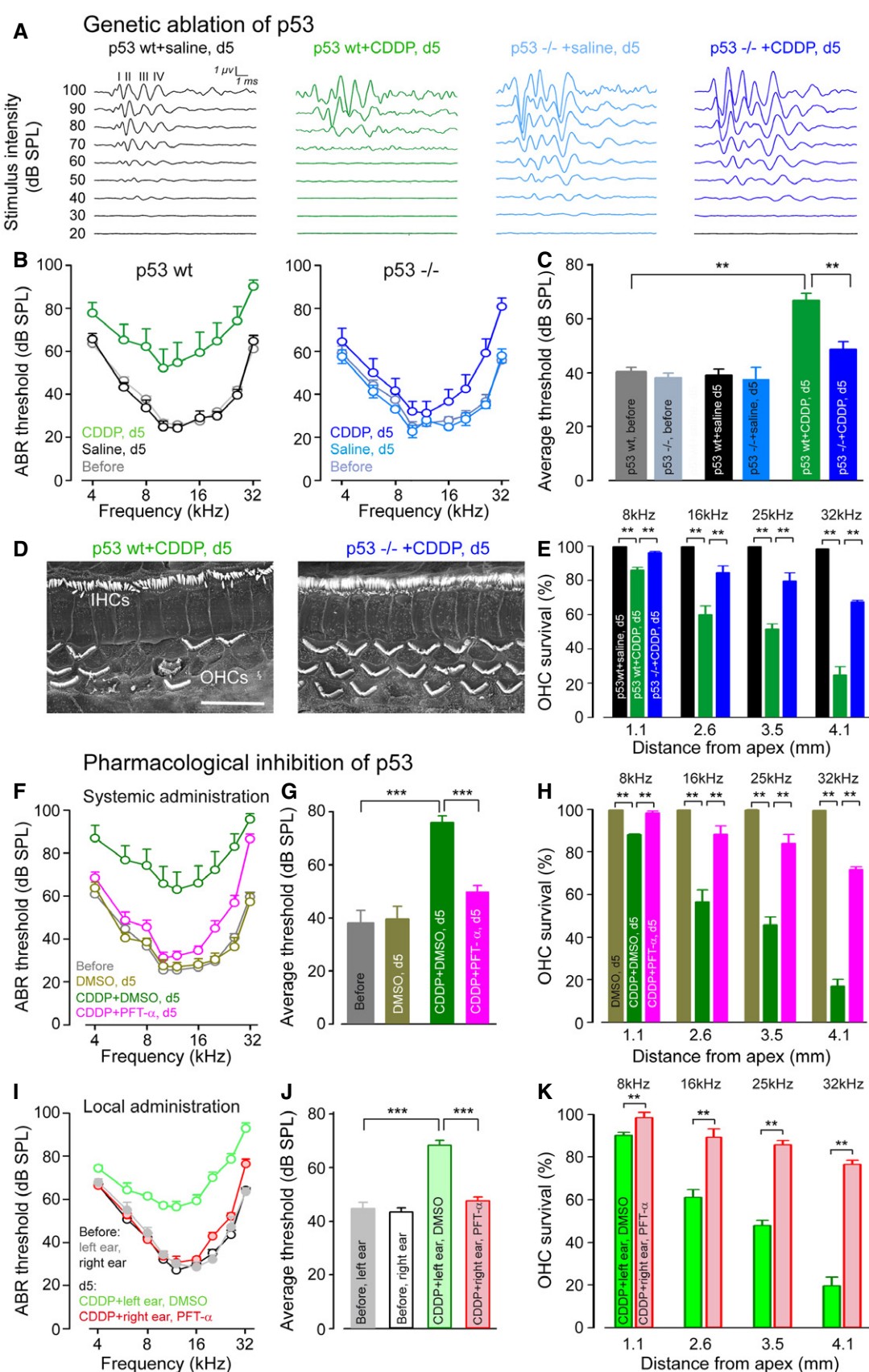

**Figure 5.**

**Figure 5.  Genetic and pharmacological deletion of p53 prevents loss of hearing and hair cells in adult mice.**

A  Representative auditory brainstem response (ABR) waveforms evoked by 16 kHz tone bursts in p53wt mice treated with saline (black plot) or CDDP (green plot), and p53$^{-/-}$ mice treated with saline (light blue plot) or CDDP (dark blue plot) for 5 days.

B  ABR thresholds recorded in p53wt mice before (gray plot) and after 5 days of saline (black plot) or CDDP treatments (green plot), and ABR thresholds recorded in p53$^{-/-}$ mice before (light gray plot), and after 5 days of saline (light blue plot) or CDDP treatment (dark blue plot). Saline-treated group: $n = 7$; CDDP-treated group: $n = 12$.

C  Mean ABR threshold from 4 kHz to 32 kHz derived from (B). One-way ANOVA test followed by *post hoc* Tukey's test (**$P \leq 0.008$; p53wt + CDDP, d5 versus p53wt, before or p53wt + CDDP, d5 versus p53$^{-/-}$ + CDDP, d5).

D  Representative scanning electron microscopy micrographs showing the basal regions of cochleae from CDDP-treated p53wt and p53$^{-/-}$ mice after 5 days. Scale bar = 15 μm.

E  Cytocochleograms representing the percentage of surviving hair cells in four cochlear regions located at 1.1, 2.6, 3.5, or 4.1 mm from the cochlear apex provided from saline-treated p53wt mice (black bars), CDDP-treated p53wt mice (green bars), or p53$^{-/-}$ mice (blue bars), after 5 days ($n = 5$ per group). Kruskal–Wallis test followed by *post hoc* Dunn's test (**$P \leq 0.008$, p53wt + CDDP, d5 versus p53wt + saline, d5 or p53wt + CDDP, d5 versus p53$^{-/-}$ + CDDP, d5).

F  ABR thresholds from p53wt mice recorded prior to (gray plot) or after 5 days systemic treatment with: DMSO (yellow plot), CDDP + DMSO (green plot), CDDP + PFT-α (pink plot). DMSO-treated group: $n = 7$; CDDP + DMSO-treated group: $n = 12$; CDDP + PFT-α-treated group: $n = 12$.

G  Mean ABR threshold from 4 to 32 kHz derived from (F). One-way ANOVA test followed by *post hoc* Tukey's test (***$P \leq 0.0005$; CDDP + DMSO, d5 versus before or CDDP + DMSO, d5 versus CDDP + PFT-α, d5).

H  Cytocochleograms representing the percentage of surviving hair cells in four cochlear regions located at 1.1, 2.6, 3.5, or 4.1 mm from the cochlear apical end taken from p53wt mice treated with: DMSO (yellow bars), CDDP + DMSO (green bars), CDDP + PFT-α (pink bars), after 5 days ($n = 5$ per group). Kruskal–Wallis test followed by *post hoc* Dunn's test (**$P \leq 0.008$; CDDP + DMSO, d5 versus DMSO, d5 or CDDP + DMSO, d5 versus CDDP + PFT-α, d5).

I  ABR thresholds from p53wt mice recorded prior to (gray plot and black plot for left and right ear, respectively) or after 5 days of systemic treatment with CDDP plus intratympanic injection of DMSO into the left ear (green plot) and of PFT-α into the right ear (pink plot). $n = 14$ per group.

J  Mean ABR threshold from 4 to 32 kHz derived from (I) ($n = 14$). One-way ANOVA test followed by *post hoc* Tukey's test (***$P = 0.0003$; CDDP + left ear, DMSO versus before, left ear or CDDP + left ear, DMSO versus CDDP + right ear, PFT-α).

K  Cytocochleograms representing the percentage of surviving hair cells in four cochlear regions located at 1.1, 2.6, 3.5, or 4.1 mm from the cochlear apical end taken from the DMSO-treated left ear (green bars) and PFT-α-treated right ear (pink bars) of the same CDDP-treated p53wt mice after 5 days ($n = 5$ per group). Kruskal–Wallis test followed by *post hoc* Dunn's test (**$P \leq 0.008$; CDDP + right ear, PFT-α versus CDDP + left ear, DMSO).

Data information: Data are expressed as mean ± SEM.

stained tumor sections. In accordance with this hypothesis, we observed a significantly reduced vascular area in the tumor periphery of *TP53*-mutant HBCx-14 but not *TP53* wt HBCx-90 tumors at 1 week after the end of combined therapy (d21, Fig 7A and B). Consistent with these results, combined therapy-treated *TP53*-mutant tumors displayed high levels of cleaved caspase-3-positive cells in vimentin labeled stromal compartment at 4 days after the end of treatments (d18), when compared with CDDP alone (Fig 7C).

**p53 pathway functional integrity assessments**

To understand how p53 inhibition potentiated the anticancer effect of CDDP in the p53 mutated cancer xenograft model, we assessed the accumulation of p53 and its downstream effector p21 in these two HBCx models. As seen in Fig 7D, treatment of mice with CDDP resulted in both reduced stabilization of p53 and accumulation of p21 in HBCx-14 (*TP53*-mutant) compared to HBCx-90 (*TP53* wt) tumors. These results suggest that the *TP53*-mutant HBCx-14 tumors retain some p53 residual transactivation activity as previously reported in sarcoma cell line and squamous cell carcinoma resected from the oral cavity of patients (Pospisilova *et al*, 2004; Perrone *et al*, 2010).

**Reversible p53 inhibition reduced the CDDP-induced autophagy selectively in the *TP53*-mutant HBCx-14 tumors**

Previous studies revealed that in response to genotoxic stress, p53-deficient tumor cells may arrest in the S and G2 phases via Chk1 activation to allow time for DNA repair and that Chk1 inhibitors selectively potentiate the effects of DNA-damaging agents, such as chemotherapy or radiation, in *TP53*-mutated cancer cells (Zhao *et al*, 2002; Ma *et al*, 2011). Based on these findings, we tested the probability of an abrogation of CDDP-induced Chk1 activation with

the CDDP + PFT-α combination in the *TP53*-mutant HBCx tumors. As expected, we found that CDDP led to an increase in Chk1 phosphorylation (p-Chk1) selectively in *TP53*-mutant HBCx-14 tumors (Fig 7E). However, this CDDP-induced Chk1 phosphorylation was not modified with PFT-α (Fig 7E).

Therefore, we investigated a potential cytoprotective role of autophagy as previously proposed to occur during anticancer therapy (Harhaji-Trajkovic *et al*, 2009; Chaachouay *et al*, 2011). We found that *TP53*-mutant tumors displayed lower basal and drug-induced (PFT-α alone or CDDP alone) levels of Beclin 1, a major player in the autophagic initiation process (Cao & Klionsky, 2007) compared to the *TP53* wt tumors (Fig 7E). In addition, combination of PFT-α and CDDP significantly attenuated Beclin 1 expression selectively in the *TP53*-mutant tumors (Fig 7E and G). Similar results were obtained in both tumor models with Western blotting analysis for LC3-II (Fig 7F and H), an autophagosomal marker, and Rab7 (Fig 7F and I), a small GTP-binding protein that has a role in maturation of late autophagic vacuoles (Jager *et al*, 2004). Taken together, our results suggest that selective suppression of autophagy in *TP53*-mutant tumors by the combination of CDDP and PFT-α may at least in part account for the observed enhanced effect.

# Discussion

The data presented here, from both *in vitro* and *in vivo* investigations, provide evidence in favor of: (i) activation of the ATM-Chk2-p53 pathway by genotoxic stress being the major determinant of CDDP ototoxicity; (ii) targeting this signaling pathway through genetic or pharmacological ablation of p53 attenuating cochlear hair cell death, and preserving hearing function during CDDP treatment; (iii) efficient hearing protection being achievable through local

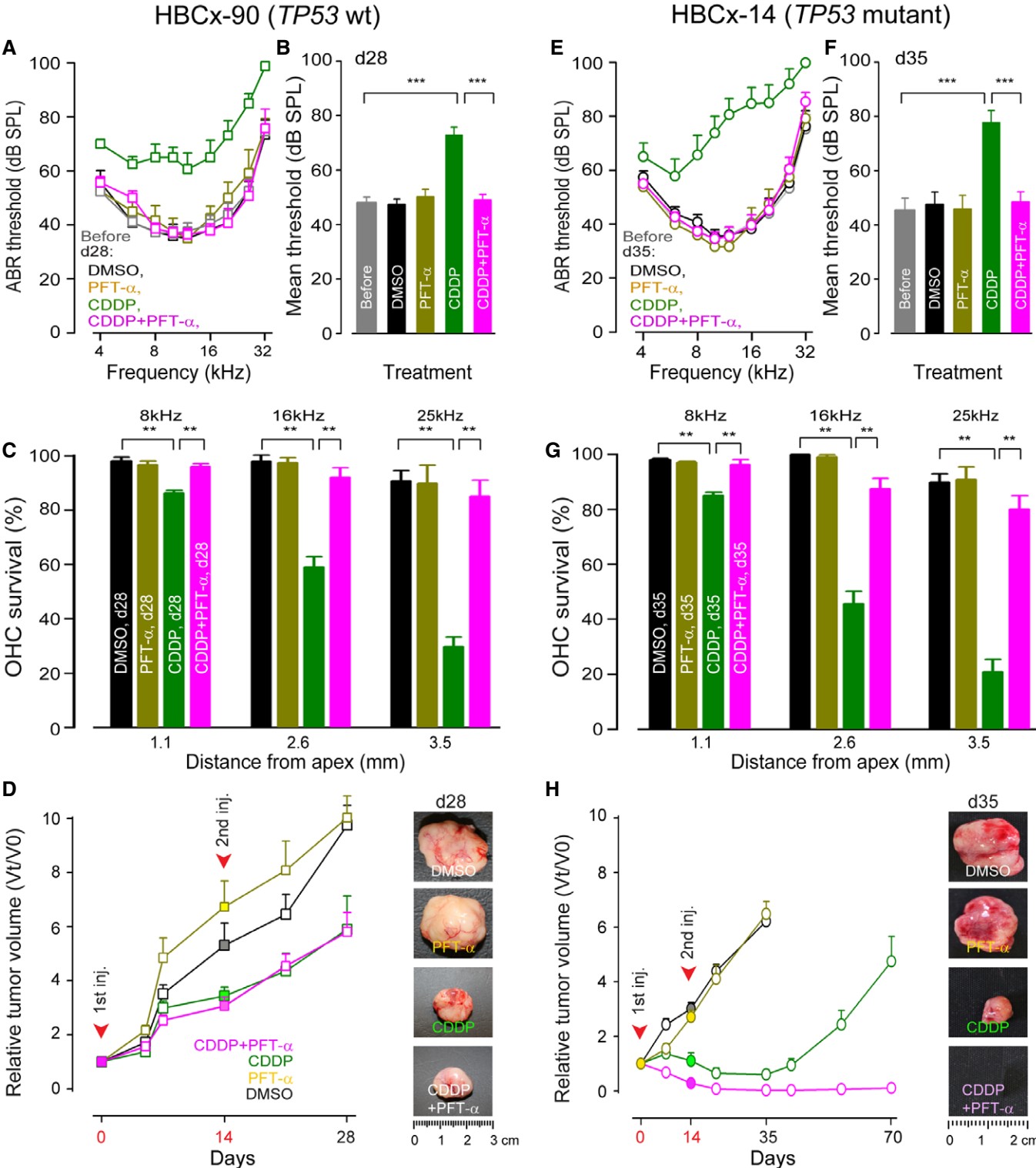

**Figure 6.**

intratympanic injection of PFT-α, a suitable method for clinical practice in any type of CDDP-based cancer therapy; and (iv) systemic administration of CDDP, combined with PFT-α, efficiently protecting against hearing loss without compromising chemotherapeutic efficacy, and even sensitizing *TP53*-mutant tumors to CDDP.

While previous studies have allowed a general understanding of the DNA damage response in cycling and cancer cells, response pathways in post-mitotic cells have been poorly studied, especially in the sensory cells of the cochlea. By using two well-characterized DNA damage markers (γH2AX and 53BP1), we have discovered that

◀

**Figure 6. PFT-α protects cochlea from ototoxicity without compromising and even enhancing CDDP anti-tumor efficacy in patient-derived breast cancer xenograft mice.**

A   ABR thresholds recorded prior to (gray plot) or at day 28 in HBCx-90 (*TP53* wt)-bearing mice received systemic treatment with: DMSO (black plot), PFT-α (yellow plot), CDDP (green plot), CDDP + PFT-α (pink plot). Before: *n* = 20; DMSO: *n* = 5; PFT-α: *n* = 5; CDDP: *n* = 5; CDDP + PFT-α: *n* = 5.

B   Mean ABR threshold from 4 to 32 kHz derived from (A). One-way ANOVA test followed by *post hoc* Tukey's test (***$P$ = 0.0002; CDDP versus before, CDDP versus CDDP + PFT-α).

C   Cytocochleograms representing the percentage of surviving hair cells in three cochlear regions located at 1.1, 2.6, and 3.5 mm from the cochlear apical end. These cochleae (*n* = 5 per group) were collected at 28 days from the beginning of treatment with DMSO (black bars), PFT-α (yellow bars), CDDP (green bars), or CDDP + PFT-α (pink bars). Kruskal–Wallis test followed by *post hoc* Dunn's test (**$P$ ≤ 0.007; CDDP, d28 versus DMSO, d28 or CDDP, d28 versus CDDP + PFT-α, d28).

D   Tumor growth curves alongside images of dissected tumors collected on day 28. Note the partial inhibition of growth in CDDP and CDDP + PFT-α-treated mice. 0, 5, 7 and 14 days: DMSO- or PFT-α-treated group: *n* = 10; CDDP- or CDDP + PFT-α-treated group: *n* = 11. 21 days: DMSO- or PFT-α-treated group: *n* = 7; CDDP- or CDDP + PFT-α-treated group: *n* = 8. 28 days: *n* = 5 per group.

E   ABR thresholds recorded prior to (gray plot) or at day 35 in HBCx-14 (*TP53*-mutant)-bearing mice received systemic treatment with DMSO (black plot), PFT-α (yellow plot), CDDP (green plot), CDDP + PFT-α (pink plot). Before: *n* = 30; DMSO: *n* = 5; PFT-α: *n* = 5; CDDP: *n* = 10; CDDP + PFT-α: *n* = 10.

F   Mean ABR threshold from 4 kHz to 32 kHz derived from (E). One-way ANOVA test followed by *post hoc* Tukey's test (***$P$ = 0.0003; CDDP versus before or CDDP versus CDDP + PFT-α).

G   Cytocochleograms representing the percentage of surviving hair cells in three cochlear regions located at 1.1, 2.6, and 3.5 mm from the cochlear apical end. These cochleae (*n* = 5 per group) were collected at 35 days from the beginning of treatment with DMSO (black bars), PFT-α (yellow bars), CDDP (green bars), or CDDP + PFT-α (pink bars). Kruskal–Wallis test followed by *post hoc* Dunn's test (**$P$ ≤ 0.008; CDDP, d35 versus DMSO, d35 or CDDP, d35 versus CDDP + PFT-α, d35).

H   Tumor growth curves alongside images of dissected tumors collected on day 35. Note the complete disappearance of tumors at day 35 and the significantly reduced tumor regrowth up to day 70 in CDDP + PFT-α-treated mice. 0, 7 and 14 days: DMSO- or PFT-α-treated group: *n* = 11; CDDP-treated group: *n* = 16; CDDP + PFT-α-treated group: *n* = 17. 21 days: DMSO- or PFT-α-treated group: *n* = 8; CDDP-treated group: *n* = 13; CDDP + PFT-α-treated group: *n* = 14. 35 days: DMSO- or PFT-α-treated group: *n* = 5; CDDP-treated group: *n* = 10; CDDP + PFT-α-treated group: *n* = 10. 42, 56 and 70 days: CDDP- or CDDP + PFT-α-treated group *n* = 5 per group.

Data information: Data are expressed as mean ± SEM.

in response to CDDP intoxication, post-mitotic cochlear sensory hair cells can recognize and manage genomic lesions using DNA damage response pathways. However, an inefficient or delayed capacity to repair DNA damage specifically in OHCs may explain the larger accumulation of both γH2AX and 53BP1 foci in their nuclei, possibly increasing their vulnerability to genotoxic stress. The observation that a substantial fraction of 53BP1 foci co-localize with both γH2AX foci and phosphorylated ATM indicates the presence of double-strand breaks (Bonner *et al*, 2008; Knobel *et al*, 2011). However, since only 20–30% of γH2AX foci co-localized with 53BP1, we believe that other types of DNA damage, such as inter- and intra-strand cross-links, may occur in CDDP-treated cochlear cells, as previously reported in cancer cells (Ang *et al*, 2010; Woods & Turchi, 2013).

ATM and ATR, two early sensors of DNA damage, play a crucial role in cell fate decision upon DNA damage and also contribute to the regulation of apoptosis in response to double-strand breaks. Although no evidence exists for the involvement of the ATR-Chk1 pathway in CDDP-induced DNA damage responses in cochlear cells, the ATM-Chk2-p53 signaling pathway is clearly activated in CDDP-intoxicated auditory sensory hair cells. The inhibition of ATM activation in our assay demonstrated a dual and opposing role of ATM in the repair of double-strand breaks in IHCs versus OHCs, leading to contrasting cell behavior, that is, OHC survival and IHC loss. In mildly damaged cells such as the IHCs following CDDP, few γH2AX and 53BP1 foci could be recovered through DNA damage repair mechanisms. In these cells, the inhibition of ATM would increase the number of unrepaired DNA damage foci and subsequently lead to IHC loss. By contrast, in OHCs displaying severe DNA damage as illustrated by higher levels of γH2AX and 53BP1, p-ATM would result in the activation of the cell death pathway and OHC loss. Thus, the inhibition of ATM protected these cells from apoptosis. Our results are consistent with previous reports showing opposite roles of ATM activation in radiation-induced apoptosis within the nervous system (Chong *et al*, 2000; Lee & McKinnon, 2000). In our study, the ability of ATM to favor double-strand break repair in IHCs may arise

from its action on Artemis, by promoting non-homologous end joining (Riballo *et al*, 2004). Due to the unfavorable effects of inhibiting ATM on the IHCs despite its favorable effect on the OHCs, we decided to probe the signaling cascade downstream of ATM.

One key substrate of ATM is the major tumor suppressor gene p53, which plays important roles in both apoptosis and DNA damage repair pathways, each pivotal for genomic stability. The onset of either p53-dependent DNA repair or apoptosis is believed to be determined by the level of accumulated damaged DNA (Offer *et al*, 2002). In our experimental paradigm, we showed that p53 accumulation and activation play a critical role in CDDP-induced IHC and OHC death, since genetic deletion or pharmacological inhibition of p53 attenuated the loss of both types of hair cell, thus preserving hearing during CDDP treatments. These results are in agreement with previous studies showing the crucial role of p53 in CDDP-induced normal tissue injuries, such as renal cell apoptosis and kidney injury (Pabla *et al*, 2008; Sprowl *et al*, 2014), as well as dorsal root ganglion neuron toxicity (Podratz *et al*, 2011). Together with previous reports demonstrating the protective effect of PFT-α against CDDP-induced cochlear and vestibular hair cell apoptosis *in vitro* (Zhang *et al*, 2003) and rat kidney proximal tubular cell apoptosis in culture (Jiang *et al*, 2004), we suggest that p53 suppression could decrease damage and promote recovery of injuries to normal tissue caused by anticancer treatments.

Having established the powerful otoprotective effect of systemic administration of PFT-α, we set out to test the efficiency of its local administration through an intratympanic approach under the guidance of customized microendoscopy. This clinically suitable technique (Bozzato *et al*, 2014) allows a maximized drug concentration in the inner ear with minimized systemic exposure (El Kechai *et al*, 2015). Our results support the local application of PFT-α as an attractive strategy to preserve hearing for any CDDP-based human cancer therapies.

Finally, we examined whether systemic administration of PFT-α interferes with the anticancer effects of CDDP, thus assessing its feasibility in human cancer therapy. Despite over half of all human cancers lacking functional p53 (Soussi & Wiman, 2007), the

**Figure 7.  PFT-α enhances CDDP-induced anti-angiogenesis and suppresses autophagy selectively in *TP53*-mutant tumors.**

A     Representative confocal images of microvessels in transversal tumor sections from HBCx-14 (*TP53*-mutant) tumors treated with either DMSO, CDDP, or a combination of CDDP and PFT-α. The sections were immunolabeled for CD31 (red) and viewed with a 20× objective. The basal-like breast cancer cells were immunolabeled in green with an antibody against cytokeratin 5 and 8. Upper right panel is a 2D projection from the white boxed area in upper left panel. The red area corresponds to CD31-labeled endothelial area, and the white area represents the lumen area. The white line shows the vessel perimeter. Scale bars = left panels, 50 μm; right panels, 35 μm. The tumor samples were collected at day 21.

B     Histograms representing the percentage of vascular area calculated using 2D reconstruction image analysis and the formula: vessel/vascular area = area of CD31-positive objects + lumen area per field area × 100%. Both HBCx-14 (*TP53*-mutant) and HBCx-90 (*TP53* wt) tumors from the different treated groups were collected at day 21 (n = 4 sections/tumor and 3–4 tumors/group). Data are expressed as mean ± SEM. Kruskal–Wallis test followed by *post hoc* Dunn's test (***$P \le 0.0004$; CDDP versus DMSO or CDDP versus CDDP + PFT-α).

C     Representative confocal images of cleaved caspase-3-positive cells (red) in the transversal tumor sections and viewed with a 20× objective. The stromal compartments were immunolabeled in green with an antibody against vimentin. HBCx-14 (*TP53*-mutant) tumors from the different treated groups were collected at day 18. Scale bar = 50 μm. c-cas-3: cleaved caspase-3.

D     Representative Western blot analysis using antibodies against β-actin, p53, and p21 in tumor extracts from HBCx-14 (*TP53*-mutant) and HBCx-90 (*TP53* wt) tumors treated with either DMSO or CDDP and collected at day 18. Note the higher CDDP-induced increase in p53 and p21 expression in *TP53* wt HBCx-90 tumors.

E, F  Representative Western blot analysis using antibodies against β-actin, p-Chk1, Beclin 1, LC3-I/II, and Rab7 in tumor extracts from HBCx-14 (*TP53*-mutant) and HBCx-90 (*TP53* wt) tumors treated with either DMSO, PFT-α, CDDP, or a combination of CDDP and PFT-α. The tumor samples were collected at day 18.

G–I   Histograms representing the levels of Beclin 1, LC3-II, and Rab7 in HBCx-14 and HBCx-90 tumors treated with the different regimens (n = 3–4 tumors per group, all experiments were performed in triplicate). Actin served as a loading control. Data are expressed as mean ± SEM. One-way ANOVA test followed by *post hoc* Tukey's test (G: *$P = 0.03$, **$P = 0.006$, ***$P \le 0.0004$; H: *$P = 0.02$, ***$P = 0.0005$; I: *$P = 0.03$, ***$P \le 0.0006$; CDDP versus DMSO or CDDP versus CDDP + PFT-α).

presence of p53 within cancer stromal cells (endothelial cells or fibroblasts) may play an important role in tumor growth and tumor response to cytotoxic anticancer agents (Bar *et al*, 2010; Shtraizent *et al*, 2016). Indeed, p53 mutations are rare in tumor-associated stromal cells (Polyak *et al*, 2009).

The tumor xenograft models that we used in the present study were obtained from directly transplanting patient-derived tumor fragment into mice. In contrast to cell line-derived xenografts, these human-to-mice tumor xenografts maintain the cell differentiation, morphology, and drug response properties of the original patient tumors (Marangoni *et al*, 2007). Using two xenograft models of triple-negative human breast cancer with either wt or mutant p53 status, we demonstrated that reversible inhibition of p53 protects the hearing function, without compromising the anti-tumor efficacy of CDDP and even sensitizing *TP53*-mutant tumors to CDDP.

An interesting explanation for this enhanced effect may be the activation of p53 playing a protective role in tumor endothelium under genotoxic stress conditions. Indeed, previous reports have shown a greater effectiveness of antiangiogenic scheduling of chemotherapy in p53-null mice (Browder *et al*, 2000) and that repression of p53 in tumor stroma sensitizes p53-deficient tumors to radiotherapy and cyclophosphamide chemotherapy (Burdelya *et al*, 2006). Alternatively, inhibiting the protective role of p53 in inducing growth arrest and DNA repair would increase the risk of mitotic catastrophe in tumor stromal cells (Bunz *et al*, 1999; Komarova *et al*, 2004). In this scenario, only proliferating cell populations would be affected, which could explain why inhibition of p53 did not cause collapse of vascular endothelia in normal mouse tissues to the same extent as tumors. Although high levels of cleaved caspase-3-positive cells were detected in stromal compartment of combined therapy-treated *TP53*-mutant tumors, we failed to see an enhancement of anti-tumor efficacy of CDDP with PFT-α in *TP53* wt tumor (HBCx-90). Therefore, we cannot exclude an impact of other mechanisms, such as the behavior of the p53-mutant tumor itself or its interaction with the tumor-associated microenvironment, as has been suggested in previous reports (Yu *et al*, 2002; Klemm & Joyce, 2015).

Although the precise relationship between reversible p53 inhibition and the sensitization of the *TP53*-mutant cancer to CDDP needs to be fully disentangled by further in-depth investigations, our results suggest that suppression of autophagy at least in part accounts for this sensitization. Consistent with a previous report showing that p53 mutants inhibit autophagy as a function of their cytoplasmic localization (Morselli *et al*, 2008), we found lower levels of several autophagy-related proteins, both basal and after CDDP intoxication, in *TP53* mutants than in *TP53* wt. In addition, we observed a selective and efficient PFT-α induced suppression of CDDP-induced autophagy in *TP53*-mutant tumors. Based on these results and those of others in cancer cells showing the protective effect of autophagy activation from CDDP or 5-FU toxicity (Harhaji-Trajkovic *et al*, 2009; Guo *et al*, 2014), we suggest that the suppression of autophagy through reversible p53 inhibition in these *TP53*-mutant tumors may account in part for their sensitization to CDDP.

The data presented in this study advance our understanding of DNA damage responses within cochlear tissue and reveal key roles played by ATM-Chk2-mediated activation of p53 in CDDP ototoxicity. More importantly, our results represent a proof of concept that reversible pharmacological suppression of p53 through systemic or local application PFT-α protects auditory function without compromising the chemotherapeutic efficacy of systemically administered CDDP, and even sensitizes *TP53*-mutant tumors to CDDP. They thereby provide a strong rationale for the clinical development of PFT-α for use in combination with CDDP-based human cancer therapy.

## Materials and Methods

### Animals

Neonate Swiss mice were purchased from Janvier Laboratories (Le Genest Saint Isle). p53$^{-/-}$ mice back-crossed into the isogenic 129sv background were a gift from the Institute of Molecular Genetics of Montpellier, but were originally obtained from Transgenesis, Archiving and Animal Models Centre (see: http://transgenose.cnrs-orleans.fr/taam/cdta.php). Swiss nude mice bearing xenografts of triple-negative human breast cancer with either mutant (HBCx-14) (Marangoni *et al*, 2007) or wt (HBCx-90) p53 status were purchased from the Institute Curie (Paris, France). Mice were housed in facilities accredited by the French Ministry of Agriculture and Forestry (B-34 172 36—March 11, 2010). Experiments were carried out in accordance with both the European Communities Council Directive of 24 November 1986 (86/609/EEC) and French Ethical Committee (agreements C75-05-18 and 01476.02), regarding the care and use of animals for experimental procedures.

### Drug preparation

CDDP was purchased from Sigma. The specific inhibitors of ATM (2-(4-morpholinyl)-6-(1-thianthrenyl)-4H-pyran-4-one, KU55933, Tocris Bioscience, #3544) and p53 (pifithrin-α, PFT-α, Tocris Bioscience, #3843) were provided by Tocris Bioscience.

For *in vitro* experiments, CDDP was freshly prepared at 100 mM in pure water and diluted in culture medium to final concentrations of 0, 5, 7.5, 10, and 20 μM, within the range commonly used for *in vitro* studies (Pabla *et al*, 2008; Oh *et al*, 2011). KU55933 and PFT-α were dissolved in 100% DMSO and diluted in culture medium to a final concentration of 10 and 100 μM in 0.2% DMSO, respectively. The chosen concentration of KU55933 was based on our preliminary evaluation of its dose–response effect on CDDP-induced γH2AX expression in CDDP-treated whole cochleae (day 1). The final concentration of PFT-α was extrapolated from a previous *in vitro* study (Zhang *et al*, 2003).

For *in vivo* experiments, CDDP was freshly prepared at 0.5 mg/ml in saline and injected intraperitoneally (IP) into tumor-free p53$^{-/-}$ and p53wt mice at a dose of 16 mg/kg and into tumor-bearing Swiss nude mice at a dose of 14 mg/kg. PFT-α was dissolved in DMSO and injected IP into mice at a dose of 2.2 mg/kg in 0.4% DMSO. This dose was extrapolated from previous *in vivo* studies (Liu *et al*, 2006; Guan *et al*, 2013). For intratympanic injection, the PFT-α was dissolved in 0.2% DMSO at a concentration of 2 mM. The chosen concentration of PFT-α was based on our preliminary evaluation of its dose–response protective effect on CDDP-induced hearing loss (day 5).

## *In vitro* protocols

### *Organ of Corti, whole cochlea, and cochlear slice cultures*

Mouse whole cochleae, organ of Corti explants, and cochlear slices (Fig EV1A and B) were collected from postnatal day 3 mice and prepared according to the procedures described previously (Wang *et al*, 2003b; Jia *et al*, 2016). The advantage in using cochlear slices is their preservation of the normal cochlear 3D architecture (Fig EV1B). The whole cochleae were kept in suspension and the organ of Corti explants and cochlear slices in adherent conditions in a 6-well culture plate containing 2 ml/well of culture medium. Culture medium consisted of Dulbecco's modified Eagle's medium/nutrient mixture F-12 (DMEM/F-12) containing 2 mM L-glutamine, N-2 complement at 1×, insulin transferrin selenium at 1× (Gibco, Life technologies), 8.25 mM D-glucose, and 30 U/ml penicillin (Sigma).

### *Pharmacological interventions*

Cochlear samples were exposed to either culture medium alone or medium containing either KU55933 or PFT-α, for 24 h in a humidified incubator (37°C, 5% $CO_2$). The culture medium was then replaced with fresh medium containing CDDP with or without KU55933 or PFT-α for 6, 12 h, 1, 2, or 3 days. For 5-day experiments, explants were further grown for 2 additional days in culture medium either alone or with inhibitors. All control explants and slices were maintained in culture medium either with or without 0.2% DMSO for 6, 12 h, 1, 2, 3, or 5 days and were run concurrently with experimental cultures.

### *Cellular localization of DNA damage response proteins and apoptotic markers*

Immunocytochemistry was employed to localize DNA damage response proteins and apoptotic markers in cultured organ of Corti and cochlear slices using monoclonal mouse antibodies against phospho-ATM (1/100, Ser1981, Merck Millipore, #05-740), phospho-H2AX (1/500, Ser139, Thermo Fisher Scientific, #PA5-35464) and polyclonal rabbit antibodies against 53BP1 (1/100, Novus Biologicals, #NB100-305), and phospho-Chk2 (1/100, Thr68, Cell Signaling Technology, #2661). A polyclonal rabbit antibody against myosin 7A (1/200, Proteus Biosciences Inc, #25-6790) and a mouse monoclonal antibody against neurofilament (NF200, 1/600, Sigma-Aldrich, #N0142) were used to label hair cells and spiral ganglion neurons, respectively. All secondary antibodies were used at a dilution of 1/1,000. This included donkey anti-mouse and anti-rabbit IgG conjugated to Alexa 488, 594 or 647 (Thermo Fisher Scientific, #A-21202, #A-21206, #A-10037, #A-10042, #A-31571, #A-31573). The Hoechst 33342 dye (0.002% wt:vol in PBS 1×, Thermo Fisher Scientific, #62249) was used to stain chromatin. Annexin V conjugated to FITC was used to detect the cell surface localization of phosphatidylserine (ApoScreen kit, CellLab, Beckman Coulter, #731718). The TUNEL kit (DeadEnd™ fluorometric TUNEL System, Promega, #G3250) was used to identify apoptotic DNA fragmentation. Fluorescent tags were visualized using a confocal microscope (LSM 5 Live Duo, Zeiss). No fluorescent signal was detected in control specimens without primary antibodies.

### *Counting sensory hair cells, DNA damage foci, and spiral ganglion neurons*

Counting sensory hair cells (5 cochleae per condition and per time point) and evaluation of DNA damage foci (50 nuclei per condition and per time point) were performed from cultured organs of Corti (length 6 mm; Henry *et al*, 2000). Due to the known resistance of apical turn hair cells to CDDP cytotoxicity, cell and foci counts were assessed over a 1.5 mm length of the cochlear duct at the basal turn (4–5.5 mm from the apex, Fig EV4). Hair cell counting was performed using standard techniques (Wang *et al*, 2003b). The number of foci (i.e. γH2AX, 53BP1 or p-ATM) per nucleus, the total volume of γH2AX foci per nucleus, and cells positive for p-Chk2 were computed using algorithms with MATLAB custom-made software (MathWorks Company) that allow 3D rendering and visualization of "isosurfaces" enveloping all pixel clusters with intensities greater than a user-defined criterion value in each corresponding image channel. Co-localized nuclei/foci or foci/foci were visualized and quantified considering the 3D intersection between respective isosurface volumes. Measurements of spiral ganglion neuron density were performed in the basal cochlear turn of NF200 stained cultured cochlear slices using a 20× microscope objective (Fig EV1C). Spiral ganglion neuron counts were assessed within a 3,600 μm$^2$ area delimited by a calibrated ocular grid. A total of two to three slices per cochlea and five cochleae per condition were used for spiral ganglion neuron quantification. All experiments were performed in triplicate.

### *Expression levels of DNA damage response proteins and apoptosis*

Whole cochlea homogenates were prepared in Laemmli sample buffer. Blots were incubated with mouse monoclonal antibodies against phospho-Chk1 (1/1,000, Ser345, Cell Signaling Technology, #2348), p53 (1/1,500, Cell Signaling Technology, #2524), and β-actin (1/10,000, Sigma-Aldrich, #A1978), and with polyclonal rabbit antibodies against phospho-H2AX (1/1500, Ser139, Cell Signaling Technology, #2577), phospho-p53 (1/1500, Ser15, Cell Signaling Technology, #9289), and cleaved caspase-3 (1/1,000, Asp175, Cell Signaling Technology, #9661), prior to incubation with horseradish peroxidase-conjugated secondary antibodies (Jackson ImmunoResearch Laboratories, #115-035-144 and 111-035-144). Western blot analysis required six cochleae per condition and per time point. All experiments were performed in triplicate.

As a specific antibody against ATR was not commercially available for cochlear tissue analysis, we used PCR to evaluate the expression of ATR and its substrate, Chk1. mRNA was extracted using a RiboPure kit (Ambion). cDNA synthesis and conventional PCR were performed using the Long Range 2Step RT–PCR kit (Qiagen), according to the manufacturer's instructions. The following primers were used: mATR forward: 5′-GTT GGC CAG TGC TAC TCC AGA A-3′; mATR reverse: 5′-GCA TGT GGC AGG ATG GAG TT-3′; mChk1 forward: 5′-CAT GGC AGG GGT GGT TTA TCT-3′; mChk1 reverse: 5′-CCT GAC AGC TAT CAC TGG GC-3′ (Eurofins MWG Operon). PCR analysis required six cochleae per condition and per time point. All experiments were performed in triplicate.

## *In vivo* protocols

### *Genetic p53 deletion*

The construction of p53$^{-/-}$ mice has been published elsewhere (Jacks *et al*, 1994). The 8-week-old male and female experimental mice were randomly divided into four groups: (i) control saline-p53wt ($n = 7$), (ii) CDDP-p53wt ($n = 12$), (iii) control saline-p53$^{-/-}$ ($n = 7$), and iv) CDDP-p53$^{-/-}$ ($n = 12$). The control treatments

consisted of an IP injection of saline once daily for 6 days (1 ml, from day 0 to day 5). The CDDP treatment consisted of two IP injections at a dose of 8 mg/kg (one at 8:00 AM, the other at 8:00 PM on day 0) resulting in an accumulated dose of 16 mg/kg. The mice then received a daily IP injection of saline for 5 days.

*Pharmacological p53 inhibition with systemic administration of PFT-α*

Pharmacological prevention of CDDP ototoxicity with systemic administration of a p53 inhibitor was performed in p53wt mice. The 8-week-old male and female experimental mice were randomly divided into three groups: (i) control DMSO (dissolvent of PFT-α, $n = 7$), (ii) CDDP + DMSO ($n = 12$), and (iii) CDDP + PFT-α ($n = 12$). Control animals received daily IP injection of DMSO (0.4%, 1 ml) over 6 days (DMSO). CDDP animals received a DMSO injection (0.4%, 1 ml) 30 min before CDDP (16 mg/kg), followed by a daily injection of 0.4% DMSO over 5 days (CDDP + DMSO). PFT-α treatment consisted of a single injection (2.2 mg/kg) 30 min before CDDP (16 mg/kg) on day 0 that was followed by daily injection over 5 days (CDDP + PFT-α).

*Pharmacological p53 inhibition with local intratympanic injection of PFT-α*

Fourteen additional 8-week-old male and female p53wt mice were used to evaluate the efficiency of local intratympanic injection of PFT-α. Mice were anesthetized with isoflurane (4% induction, then 1.5% maintenance with $O_2$). Intratympanic injection was performed under visualization using a custom-made microendoscopy (MEED SYSTEMS, France) (Fig EV3A and B). It consisted in a single injection of PFT-α (2 mM, 10 μl) into the right ear 30 min before systemic CDDP injection (16 mg/kg) on day 0, followed by daily injection over 2 days. Each right ear PFT-α injection was preceded by an intratympanic injection of DMSO (0.2%, 10 μl) in the control left ear.

*p53 inhibition in patient-derived breast cancer xenograft mice*

The impact of p53 inhibition on the chemotherapeutic efficacy of CDDP was evaluated in two models of triple-negative patient-derived breast cancer xenografts, one with wt (HBCx-90) and the other with mutant (HBCx-14) p53 status. The p53-mutant tumor contained two mutations at exon 5 of *TP53*: p.Y163C (substitution—missense) and c.488A>G (substitution). The generation of these xenografts has been published previously (Marangoni *et al*, 2007). Briefly, tumor specimens were obtained from consenting patients during surgical resection. The tumor samples were established as xenografts by subcutaneous implantation of a tumor fragment into the interscapular fat pad of female Swiss nude mice. They were subsequently transplanted from mouse to mouse with a volume of ~15 mm³. After the tumors had grown to ~250–300 mm³, mice were randomly divided into four groups prior to treatment: (i) control DMSO (HBCx-14: $n = 11$, HBCx-90: $n = 10$), (ii) control PFT-α (HBCx-14: $n = 11$, HBCx-90: $n = 10$), (iii) CDDP (HBCx-14: $n = 16$, HBCx-90: $n = 11$), and (iv) CDDP + PFT-α (HBCx-14: $n = 17$, HBCx-90: $n = 11$). The control DMSO and PFT-α alone groups received two series of daily IP injections of DMSO (0.4%, 1 ml) or PFT-α (2.2 mg/kg) over 6 days with a 1-week interval, respectively. The CDDP treatment consisted of two IP injections at a dose of 7 mg/kg (one on day 0, the other on day 14; Fig 6D and H) resulting in an accumulated dose of 14 mg/kg. Each CDDP injection was

preceded by an IP injection of DMSO (0.4%, 1 ml) 30 min before, and followed by daily injections of DMSO over 5 days. The combined treatment of CDDP plus PFT-α consisted of a single injection with PFT-α (2.2 mg/kg) 30 min prior to CDDP (14 mg/kg) on day 0 and day 14, followed by daily injections of PFT-α over 5 days.

To monitor tumor biomarkers, three animals from each treatment regimen and each xenograft model were randomly selected and sacrificed at day 18. For histological features of tumors and angiogenesis, two *TP53* wt HBCx-90-bearing mice treated with either DMSO or PFT-α alone and three–four animals from each of the other groups were randomly selected and sacrificed at day 21. To assess hearing function and hair cell morphologies, five other animals from *TP53* wt HBCx-90-bearing mice needed to be sacrificed at day 28, and from *TP53*-mutant HBCx-14-bearing mice at day 35 due to tumor volume reaching around 3 cm³ in both HBCx-14- and HBCx-90-bearing mice treated with either DMSO or PFT-α alone. Five animals from the CDDP- and CDDP + PFT-α-treated *TP53*-mutant HBCx-14-bearing mice were kept up to day 70 to evaluate tumor recurrence.

*p53 genotyping and CDDP-induced p53 activation in vivo*

Genotyping was performed using routine PCR with the following primer sequences (forward and reverse, respectively): *TP53* (X6.5: 5′-ACAGCGTGGTGGTACCTTAT-3′ and X7: 5′-TATACTCA-GAGCCGGCCT-3′); neomycin (Neo 204: 5′-GCTATTCGGCTAT-GACTGGG-3′ and neo909: 5′-GAAGGCGATAGAAGGCGATG-3′). All primers were synthesized by Eurofins MWG Operon. PCR analysis confirmed the lack of p53 expression in p53$^{-/-}$ mice. In contrast, a clear expression was seen in wt and heterozygote mouse cochlear tissues (Fig EV3E). Levels of p53 activation were examined at 0, 2, and 5 days after CDDP injections in both p53wt and p53$^{-/-}$ mice using Western blots (Fig EV3F).

*Functional hearing assessments*

Auditory function was assessed by recording sound-evoked auditory brainstem responses. The recordings were performed prior to, and 5 days following CDDP treatment in p53wt and p53$^{-/-}$ mice, and 28 or 35 days from the beginning of CDDP treatment in *TP53* wt HBCx-90-bearing mice and *TP53*-mutant HBCx-14-bearing mice, respectively. The recordings were carried out under anesthesia with Rompun 2% (3 mg/kg) and Zoletil 50 (40 mg/kg) in a Faraday shielded anechoic soundproof cage. Rectal temperature was measured with a thermistor probe and maintained at 38.5°C ± 1 using a heated under blanket. Auditory brainstem responses were recorded from three subcutaneous needle electrodes placed on the vertex (active), on the pinna of the tested ear, and in the neck muscles (ground) of the mice. The acoustical stimuli were generated by a NI PXI-4461 signal generator (National Instruments) consisting of 10-ms tone bursts with a 1-ms rise and fall time delivered at a rate of 10/s. Sound was delivered by a JBL 075 loudspeaker (James B. Lansing Sound) in a calibrated free-field condition, positioned at 10 cm from the tested ear. Cochlear amplification (20,000) was achieved via a Grass P511 differential amplifier, averaged 1,000 times (Dell Dimensions). Intensity-amplitude functions of the ABRs were obtained at each frequency tested (4, 6.3, 8, 10, 12.5, 16, 20, 25, and 32 kHz) by varying the intensity of the tone bursts from 0 to 100 dB SPL, in 5 dB incremental steps. The ABR thresholds were defined as the minimum sound intensity necessary to elicit well-

defined and reproducible wave II. Recordings and analysis were performed blindly.

### Cochlear morphological assessments

Sensory hair cell loss was evaluated using scanning electron microscopy (Hitachi S4000). The cochleae from the different treatment groups ($n = 5$ per group) were processed and evaluated using previously reported standard techniques (Ladrech *et al*, 2007; Oh *et al*, 2011). In p53wt and p53$^{-/-}$ mice, hair cell counting was performed in four different 300-μm-long segments of the organ of Corti, centered at 1.1, 2.6, 3.5, or 4.1 mm from the cochlear apical end and corresponding to the frequencies of 8, 16, 25, and 32 kHz, respectively (Muller *et al*, 2005). In all Swiss nude mice bearing xenografts or not, a massive loss of OHCs was observed in the extreme basal cochlear region, independent of CDDP treatment. Therefore, in these xenograft mice, to evaluate specific hair cell loss induced by CDDP, the counting was restricted to three different 300 μm long segments of the organ of Corti, centered at 1.1, 2.6, 3.5 mm from the cochlear apical region.

### Tumor assessments, sample preparation

Tumor volume was determined using digital vernier caliper measurements once per week and using the formula: $V = (\pi/6) \times (d_1)^2 \times d_2$, where $d_2$ is the longest diameter of the tumor and $d_1$ is the shortest. At the different time points after the end of treatments, each tumor was harvested and cut into three pieces with one piece fresh frozen for Western blotting and the two other pieces fixed and prepared for cryosections or paraffin-embedded sections before being processed for immunofluorescence staining or immunohistochemistry analysis.

### Tumor histological and angiogenesis evaluation

Histological examination of tumors was performed on 5-μm formalin-fixed paraffin sections stained with hematoxylin and eosin safranin. These sections were then scanned using a NanoZoomer (Hamamatsu Photonics, Hamamatsu City, Japan) with a 40× objective and blind reviewed by a pathologist. Software designed by Definiens Developer 7.1. (Definiens, Munich, Germany) was used to estimate the ratio of fibrous scar area (only fibroelastotic stroma remaining)/tumor cell area (cellularity plus necrosis), for each entire tumor section (see Fig EV3C, $n = 4$ sections/tumor and 3–4 tumors/group).

Evaluation of tumor angiogenesis and cleaved caspase-3-positive cells was performed on 6-μm formalin-fixed cryostat sections. Monoclonal rat antibody against CD31 (1/100, Merck Millipore, #MAB1393) was used to evaluate tumor vascular area. A polyclonal rabbit antibody against cleaved caspase-3 (1/500, Asp175, Cell Signaling Technology, #9661) was used to identify apoptotic cells. Basal-like breast cancer cells were identified using a polyclonal rabbit antibody against cytokeratin 5 and 8 (1/100, Merck Millipore, #MAB3228) (Martin-Castillo *et al*, 2015). Non-malignant tumor stromal cells were identified using a monoclonal mouse antibody against vimentin (1/2,000, Sigma-Aldrich, #V 6630). The secondary antibodies used were goat anti-rat, anti-mouse or anti-rabbit IgG conjugated to Alexa 488 or 594 (1/3,000, 1/100, Merck Millipore, #MAB3228 and #111-001-003). The Hoechst 33342 dye (0.002% wt:vol in PBS 1×, Thermo Fisher Scientific, #62249) was used to stain chromatin. Negative controls were performed by omitting the primary antibody.

Five fields of CD31 stained sections, using a 20× microscope objective, were chosen in the most vascular part (periphery) of the tumor to estimate the sum of the area occupied by vessels, according to the conventional vascular area quantification in solid tumors (Vermeulen *et al*, 2002; Mikalsen *et al*, 2013). The vessels were identified in the images using a user-supervised algorithm developed using MATLAB programing language (image processing toolbox, MathWorks). CD31-positive pixels were first automatically detected using a thresholding method (*edge* function, threshold adjusted by the user). The lumen areas were then segmented using the *bwselect* function associated with the *imclose* function. In the resulting image, the profile of vessels appeared in red and their interior area (lumen) in white (Fig 7A, upper right). The vascular areas were then calculated using the formula: vascular area = CD31-positive area + lumen area. Only gaps in the stains wider than 1 μm were considered to be lumens and included in the estimate (Fig 7B). A total of four sections per tumors and three to four tumors per group were used for tumor quantification.

### Biomarker analysis

Western blotting was used for biomarker analysis. The functional integrity of the p53 pathway was analyzed for p53 and p21 in both xenograft models treated with either DMSO or CDDP. The effect of p53 inhibition on CDDP-induced DNA damage and autophagy was assessed using phosphorylated Chk-1 as DNA damage marker, and Beclin 1, LC3-II and Rab7, as autophagy markers in each xenograft model following the various treatment regimens. Western blotting analysis was performed using standard procedures. Antibodies used included those recognizing p21 (1/2,000, Cell Signaling Technology, #2946), p53 (1/1,500, Cell Signaling Technology, #2524), phospho-Chk1 (1/800, Ser345, Cell Signaling Technology, #2348), Beclin 1 (1/1,000, Santa Cruz Biotechnology, #sc11427), LC3-II (1/800, Cell Signaling Technology, #2775), Rab7 (1/800, Santa Cruz Biotechnology, #sc-376362), and β-actin (1/10,000, Sigma-Aldrich, #A1978). Secondary antibodies used were goat anti-mouse IgG (1/3,000, Jackson ImmunoResearch, #115-001-003) and goat anti-rabbit IgG (1/3,000, Jackson ImmunoResearch, #111-001-003). Western blot analysis required 3–4 tumors per group. All experiments were performed in triplicate.

## Statistics

Data are expressed as mean ± SEM. If conditions for a parametric test were met, the significance of the group differences was assessed with a one-way ANOVA; once the significance of the group differences ($P < 0.05$) was established, Tukey's *post hoc* tests were subsequently used for pairwise comparisons. If not, Kruskal–Wallis tests were used to assess the significance of differences among several groups; if the group differences were significant ($P < 0.05$), Dunn's tests were then used for *post hoc* comparisons between pairs of groups. Data analysis was performed using MATLAB (The Mathworks Company). For *in vivo* mouse studies, based on data from our previous reports (Wang *et al*, 2003a, 2004) or from preliminary experiments, we calculated the sample size using G*Power 3.1.9.2 to ensure adequate power of key experiments for detecting pre-specified effect sizes. The statistical test used and associated *P*-values are indicated in the legends of each figure.

**The paper explained**

**Problem**

Cisplatin (CDDP) is a highly effective and widely used chemotherapeutic agent for the treatment of different types of human tumor. Unfortunately, it has a number of dose limiting side effects including ototoxicity, which greatly hamper its chemotherapeutic value. The mechanism behind cisplatin-induced ototoxicity today remains unclear, and hearing preservation during cisplatin-based chemotherapy in patients is lacking. This study explored the molecular pathway of cisplatin ototoxicity from cellular events to the whole system level. Improving our understanding of the involved mechanisms could lead to measures to protect hearing without compromising the chemotherapeutic effect of cisplatin in patients undergoing cisplatin-based chemotherapy.

**Results**

The data presented, from both *in vitro* and *in vivo* investigations, provide evidence in favor of: (i) activation of the ATM-Chk2-p53 pathway by genotoxic stress being the major determinant of cisplatin ototoxicity; (ii) targeting this signaling pathway through genetic or pharmacological ablation of p53 attenuating cochlear sensory cell death, and preserving hearing function during cisplatin treatment; (iii) efficient hearing protection being easily achievable in clinical practice through local middle ear (intratympanic) injection of the reversible p53 inhibitor PFT-α, in any type of CDDP-based cancer therapy; (iv) systemic administration of CDDP, combined with PFT-α, efficiently protecting against hearing loss without compromising chemotherapeutic efficacy, and even sensitizing *TP53*-mutant tumors to CDDP.

**Impact**

The importance of ATM-Chk2-mediated activation of p53 in CDDP ototoxicity is highlighted. More importantly, the study represents a proof of concept that reversible pharmacological suppression of p53, through systemic or local administration of PFT-α, protects auditory function without compromising the chemotherapeutic efficacy of systemically administered CDDP, and even sensitizes *TP53*-mutant tumors to CDDP. The findings thus provide a strong rationale for the clinical development of PFT-α for use in combination with CDDP-based human cancer therapy.

Expanded View for this article is available online.

## Acknowledgements

The authors would like to thank to J. Sarniguet and K. Chebli for their assistance in the back-crossing of p53 knockout mice. Thanks are also due to L. de Plater for generation of patient-derived breast cancer xenograft models and to P. Kazmierczak for help with editing. All confocal and electron-microscopic analyses were performed at the Montpellier RIO Imaging-INM core facility. This work was supported by the Fondation de l'Avenir (Et2-675) INSERM (ENV201508) and the Fondation Gueules Cassées. The salaries of Nesrine Benkafadar and Julien Menardo were supported by the Fondation de l'Avenir and the French Ministry of Research and Technology, respectively. The manuscript has been revised by an independent scientific English language editing service (Angloscribe).

## Author contributions

JW designed the research project and supervised the experiments. JM carried out *in vitro* and *in vivo* cisplatin-induced ototoxicity experiments with assistance from FF. DD provided the patient-derived breast cancer xenograft models. NB performed the *in vivo* experiments with the cancer xenograft models. JW performed SEM evaluations and JB carried out quantification analysis. JW, J-LP, and RN wrote the manuscript which was then critically reviewed by DM who also supplied DNA damage markers.

## Conflict of interest

The authors declare that they have no conflict of interest.

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
