## [Review Process File · EMBO Molecular Medicine]

Reversible p53 inhibition prevents cisplatin ototoxicity without blocking chemotherapeutic efficacy

Nesrine Benkafadar, Julien Menardo, Jérôme Bourien, Régis Nouvian, Florence François, Didier Decaudin, Domenico Maiorano, Jean-Luc Puel, and Jing Wang

Corresponding author: Jing Wang, INSERM

Review timeline:

Submission date:	15 January 2016
Editorial Decision:	02 May 2016
Revision received:	22 August 2016
Editorial Decision:	09 September 2016
Revision received:	23 September 2016
Accepted:	26 September 2016

Transaction Report:

Editor: Céline Carret

1st Editorial Decision

02 May 2016

Thank you for the submission of your manuscript to EMBO Molecular Medicine. We have now heard back from the two referees whom we asked to evaluate your manuscript. Although the referees find the study to be of potential interest, they also raise a number of concerns that need to be addressed in the next final version of your article.

As you will see from the comments below, the two referees are positive about the study but do have very overlapping suggestions and recommendations to further improve conclusiveness and clarity as well as provide more mechanistic insights.

I will not get into experimental details, but we feel that the referees' reports are very clear and nicely detailed and we would strongly encourage you to address all issues raised as recommended.

Given these evaluations, I would like to give you the opportunity to revise your manuscript, with the understanding that the referees' concerns must be fully addressed and that acceptance of the manuscript would entail a second round of review. Please note that it is EMBO Molecular Medicine policy to allow only a single round of revision and that, as acceptance or rejection of the manuscript will depend on another round of review, your responses should be as complete as possible.

I look forward to receiving your revised manuscript.

***** Reviewer's comments *****

Referee #1 (Comments on Novelty/Model System):

SUMMARY

In this manuscript, the authors describe a strategy to prevent ototoxicity after cisplatin treatment. They report that ATM-Chk2-p53 pathway activation is a major determinant of cisplatin ototoxicity. ATM activation however has opposite effects on the two types of hair cells: outer hair cells undergo apoptosis, whereas inner hair cells appear more resistant. Removal of p53, both genetically or pharmacologically, ameliorates cochlear cell death and consequently preserves hearing. Moreover, unexpectedly, reversible inhibition of p53 prevents tumor recurrence in xenografted mice treated with cisplatin.

Overall, the paper shows that the known pathway for p53 activation after cisplatin treatment operates also in sensory hair cells which provides new insights into the origin of ototoxicity. In vivo studies convincingly demonstrate that ototoxicity can be prevented using a p53 inhibitor. However, as detailed below some of the findings described are not novel. Furthermore, the outcome of the xenograft experiments lack a mechanistic basis and do not convincingly justify the main conclusion that P53 inhibition enhances the antitumor efficacy of cisplatin treatment and inhibits tumor recurrence. Altogether, the main conclusions in this manuscript are at this stage not convincing.

CRITIQUE

1. Many experiments performed on explants in this manuscript are not novel. Zhang et al (Neuroscience, 2003) performed similar studies and showed p53-mediated hair cell loss upon cisplatin treatment which can be counteracted with the use of PFT-alpha. The authors do refer to this paper in the material and method section, but do not further discuss the findings of the Zhang et al publication. Data that repeating this previous publication can be largely moved to the supplemental section and proper reference needs to be made.
2. Unexpectedly, a synergistic response on tumor volume is observed after combined treatment of cisplatin and PFT-alpha compared to cisplatin alone, in stead of unaffected or enhanced tumor growth upon inhibition of p53.. The authors suggest that p53 repression in the tumor stroma might be responsible for this effect (page 14). This is an interesting idea, for which experimental evidence should be provided. Is this a general effect or specific for this type of xenografts? Is it representative for tumors in general? Also further mechanistic insights into the remarkable synergistic effect of PFT-alpha and cisplatin are missing.
3. Xenograft experiments are performed using p53-mutated xenografts (Marangoni et al, 2007). In these tumors, PFT-alpha does not counteract the cisplatin toxicity and surprisingly even enhances its effect. A possible alternative explanation is that the mutant p53 protein aids the survival of the tumor after cisplatin exposure and inhibition of (mutant) p53 sensitizes the xenograft for cisplatin. The general conclusion that reversible p53-inhibition enhances antitumor efficacy and inhibits tumor recurrence (title) not justified. Experiments with different xenograft models need to be performed including p53-positive and p53-null xenografts to confirm that PFT-alpha indeed does not counteract the antitumor activity of cisplatin overall.
4. Figure 6A shows representative pictures of xenografts and tumor volume after the different treatments. Combination treatment completely eliminates the tumor. In the rest of the figure, the authors perform several stainings on the different tumors, including the tumor of combination treated animals. How can the authors examine tumors that are supposed to be eliminated?

Minor points:

-To my opinion, the figures are quite crowded and it is difficult to maintain a clear overview. The authors might replace some non-essential data to the supplemental section to create a better overview. For example x projection and 3D pictures of the IF stainings.

-On page 7 is mentioned, ...both ATR and Chk1 mRNA are expressed... Genes are expressed and mRNA is produced and/or present.

-Fig 5B. The legend of p53 wt green bar is missing.

-Fig 5D. Why not use the same staining as in the rest of the figures (such as 1A)?

Referee #2 (Comments on Novelty/Model System):

As mentioned in the comments to the authors, numerous things have to be done to improve quality of the data. At least, statistical analyses have to be done correctly and additional breast cancer PDX have to be used.

Referee #2 (Remarks):

In this paper the authors nicely described new data regarding the anti-tumoral effect of p53 inhibition and its association to ototoxicity. They begin the paper showing the mechanism of ototoxicity of CDDP both in vitro and in vivo, that goes through p53 activation. Then, they studied the effect of a specific p53 inhibitor (PFTalpha or PFT) on ototoxicity induced by CDDP treatment in a patient derived breast cancer xenograft mouse model. While, PFT protects against CDDP ototoxicity, as expected, they surprisingly found that this treatment is strikingly effective on p53 negative tumor growth. While, it has been previously demonstrated that PFT protects hair cells from cisplatin ototoxicity, the effect of PFT on tumor growth is really interesting and shed light on new molecular mechanisms that can give rise to new treatments for cancer and ototoxicity.

However, there are numerous points that need to be clarified:

PFT is a p53 inhibitor and the authors showed that it could help to target p53 negative tumors. This is difficult to understand even if the authors claim: "the presence of p53 within cancer stromal cells (endothelial cells or fibroblasts) may play an important role in tumor growth". The authors have to show p53 positive cells (endothelial cells?) around the tumor. If such, how the authors can explain the absence of effect of PFT alone (figure 6D) on vascular area? In addition, how the authors explain that CD133 cancer stem cells - that should be p53 negative - are decreased in the presence of CDDP and PFT? That has to be explained.

It really seems that there is a synergic effect of CDDP and PFT on tumor growth. This needs to be explained at molecular level, since tumors are p53 negative and since PFT has no effect alone.

The authors based the entire study on one patient-derived xenograft (PDX). In order to prove that p53 inhibition is really effective to decrease p53 negative tumor size, the authors have to extend their study to other breast cancer PDX (that are numerous) and compare p53 positive and p53 negative tumors.

Finally, they also have to address the problem of metastasis since it has been recently shown that p53 deficiency enhances tumor growth at metastatic sites (Breast cancer res, 2016, Powell et al.).

Other comments:

Statistical analyses: the authors mentioned that they have done student-t-test or Wilcoxon's test. However, in the majority of the graphs there is 3 or more groups, that need to be analyzed by ANOVA followed by a post test. In addition, it is not mentioned when a parametric or non-parametric test has been used. Therefore all the statistical analyses should be explained in details and corrected for data containing 3 or more groups.

Figure 1

As cisplatin is a DNA intercalating drug it may be toxic for both hair cells and neurons. No images at the level of neurons are shown and should be added to see if cochlear cell death mechanism is ubiquitous or not.

panel C: what are the different curves? there are 2 blue and 2 red (with different intensities). This is not explained in the legend and should be added.

panel D: whole cochlear extracts?

panel E: how caspase-3 level is quantified (fold activation??)

Figure 4:

panel 4D: Differential effect of KU55933 (ATM inhibitor) in OHC and IHC. it is not clear why pATM favors DSB repair in IHC while it induces apoptosis in OHC. CDDP induced DSB that induces pATM, pCHK2 and p-P53 to finally induces apoptosis. Therefore the effect of KU55933 on IHCs is really intriguing and needs to be clarified.

In general, the results regarding a differential effect on OHCs and IHCs are not clear. The authors showed in figure 1 that CDDP is not inducing any IHCs death (and clearly there is no annexin V nor tunel positive IHCs) but they then showed on figure 4H that CDDP seems to have an effect that is restored upon PFT treatment (and mentioned it in the text p8). In addition as mentioned above, statistical analysis is not clear on this figure.

Figure 6:

Effect of PFT alone on tumor volume? (even if there is no effect on CD31 and CD133 cells).

The authors claim that CDDP +PFT treatment inhibits local tumor recurrence but they only showed one stage of recurrence (70 days) where a difference appears. They should show another stage to claim such statement.

What is the explanation for the synergic effect of CDDP and PFT on vascular area and cd133 positive cell number since PFT alone has no effect. As the authors suggest in the discussion that vascular cells and stem cells may be p53 positive it is difficult to understand the absence of effect of PFT alone.

fig 6D : typographic error for cytokeratin

Mat and met:

numerous sections need additional details:

- animals: why do the authors used swiss mice for in vitro experiments and not mice from the same background as p53-KO mice (129sv).

- Construction of p53 ko mice should be described a little (at least a good reference should be added since the authors cited a wrong ref from jackson and raymond that deals with MITF and not p53 and that is a comment on two other papers!!!)

- mice bearing xenograft: the authors should explained the rationale to used this specific p53-Rb-PTEN triple-neg breast cancer. It seems that the authors could have used many other HBCx tumors as described previously (breast cancer research, 2014, grinde et al.) or even at jackson lab where numerous tumors are commercialized.

- How many cells have been graft and where? This should be clarified.

- What is the brand of the apparatus allowing auditory function recording?

the authors mentioned: "The ABR thresholds were defined as the minimum sound intensity necessary to elicit a clearly distinguishable response ($> 0.2\mu\text{V}$)". They should clarify what does this means? For which peak? is it a blind analysis?

- Did the hair cell counts were done in the same way for p53 and PDX mice? not clear in the text

- catalog number for drugs that have been used should be added

- cultures : how the explants are maintained in 6 well plates? In suspension or in adherent conditions?

- Detailed info for antibodies should be added (catalogue number, exact name of the society : for example, molecular probes does not exists anymore, it is fisher that distributes the corresponding products).

1st Revision - authors' response

22 August 2016

ANSWERS TO THE REVIEWER'S COMMENTS

We would like to thank both reviewers for their insightful comments on the paper, as these comments led us to an improvement of the work. Our revisions reflect all reviewers' suggestions and comments. Detailed responses to reviewers are given below.

Reviewer 1**1) Reviewer comments**

In this manuscript, the authors describe a strategy to prevent ototoxicity after cisplatin treatment. They report that ATM-Chk2-p53 pathway activation is a major determinant of cisplatin ototoxicity. ATM activation however has opposite effects on the two types of hair cells: outer hair cells undergo apoptosis, whereas inner hair cells appear more resistant. Removal of p53, both genetically or pharmacologically, ameliorates cochlear cell death and consequently preserves hearing. Moreover, unexpectedly, reversible inhibition of p53 prevents tumor recurrence in xenografted mice treated with cisplatin.

Overall, the paper shows that the known pathway for p53 activation after cisplatin treatment operates also in sensory hair cells which provides new insights into the origin of ototoxicity. In vivo studies convincingly demonstrate that ototoxicity can be prevented using a p53 inhibitor.

Authors' answer

The authors would like to thank the reviewer for the careful review and the helpful advice for improving the manuscript. He found this work provides new insights into the origin of ototoxicity and in vivo studies convincingly demonstrate that ototoxicity can be prevented using a p53 inhibitor.

2) Reviewer comments

However, as detailed below some of the findings described are not novel. Furthermore, the outcome of the xenograft experiments lack a mechanistic basis and do not convincingly justify the main conclusion that P53 inhibition enhances the antitumor efficacy of cisplatin treatment and inhibits tumor recurrence. Altogether, the main conclusions in this manuscript are at this stage not convincing.

Authors' answer

We definitely agree with the reviewer that the main conclusion needs further investigations, i.e., can it be extended to other tumors or is it P53 mutated specific? Therefore we extended our study to another triple negative patient-derived breast cancer xenograft model, in which p53 status is wild type (HBCx-90). In this breast cancer xenograft (p53 wild-type), the inhibition of p53 did not affect the efficiency of CDDP. Therefore, our results suggest that the anti-tumor enhancement of CDDP through p53 inhibition operates selectively in *TP53* mutant human breast cancer. These new data are now addressed in the result section (page 10, line 15-24 and page 11, line 1-8).

We have accordingly rewritten our conclusion (page 19, line 2-9).

“The data presented in this study advance our understanding of DNA damage responses within cochlear tissue and reveal key roles played by ATM-Chk2-mediated activation of p53 in CDDP ototoxicity. More importantly, our results represent a proof-of-concept that reversible pharmacological suppression of p53 through systemic or local application PFT- α protects auditory function without compromising the chemotherapeutic efficacy of systemically administered CDDP, and even sensitizes *TP53* mutant tumors to CDDP. They thereby provide a strong rationale for the clinical development of PFT- α for use in combination with CDDP-based human cancer therapy.”

3) Reviewer comments

Many experiments performed on explants in this manuscript are not novel. Zhang et al (Neuroscience, 2003) performed similar studies and showed p53-mediated hair cell loss upon cisplatin treatment which can be counteracted with the use of PFT-alpha. The authors do refer to this paper in the material and method section, but do not further discuss the findings of the Zhang et al publication. Data that repeating this previous publication can be largely moved to the supplemental section and proper reference needs to be made.

Authors' answer

As suggested, we have removed our *in vitro* data concerning p53-mediated hair cell loss upon CDDP treatment and its counteraction with the use of PFT-a to the Expanded View data. The Zheng et al publication was discussed in the discussion section.

See Fig EV2G-H and its legend.

And discussion section: page 16, Line 11-12.

“Together with previous reports demonstrating the protective effect of PFT- α against CDDP-induced cochlear and vestibular hair cell apoptosis *in vitro* (Zhang *et al*, 2003).”

4) Reviewer comments

Unexpectedly, a synergistic response on tumor volume is observed after combined treatment of cisplatin and PFT-alpha compared to cisplatin alone, instead of unaffected or enhanced tumor growth upon inhibition of p53. The authors suggest that p53 repression in the tumor stroma might be responsible for this effect (page 14). This is an interesting idea, for which experimental evidence should be provided. Is this a general effect or specific for this type of xenografts? Is it representative for tumors in general? Also further mechanistic insights into the remarkable synergistic effect of PFT-alpha and cisplatin are missing.

Authors' answer

We agree with the reviewer and this point was taken into consideration. We have now added new experiments to identify the mechanisms underlying the enhanced effect CDDP plus PFT-a.

Our new data with immunostaining showed that high levels of cleaved caspase 3 positive cells were predominantly located in vimentin labelled stromal compartment from the combination of CDDP+ PFT-a-treated *TP53* mutant HBCx-14 tumors, when compared with CDDP alone. These results support the hypothesis that the inhibition of p53-mediated G1 cell cycle arrest and DNA damage repair with PFT-a upon CDDP genotoxicity exposure probably impedes the DNA repair and hence redirects stromal cells toward apoptosis. However, we failed to see an enhancement using the *TP53* wild-type tumor (HBCx-90). Therefore, we cannot exclude other mechanisms, such as the behavior of the p53 mutant tumor itself or its interaction with the tumor associated microenvironment as has been suggested in previous studies (Yu *et al*, 2002; Klemm & Joyce, 2015). These points are now discussed.

More importantly, our new data with western blot analysis revealed that the basal levels, as well as those induced by drugs (PFT-a alone or CDDP alone), of several autophagy proteins were lower in *TP53* mutant than *TP53* wt tumors. In addition, combination of PFT-a and CDDP significantly attenuated the expression of autophagy protein selectively in *TP53* mutant HBCx-14 tumors. These results suggest that selective suppression of autophagy in *TP53* mutant tumors with combination of CDDP and PFT-a may account in part for the enhancing effect observed.

As raised in a previous comment (2), we probed CDDP plus PFT-a effect on another triple negative patient-derived breast cancer xenograft model carrying wild type p53 (HBCx-90). No enhancement of the anti-tumor effect was observed, suggesting that the increase in the anti-tumoral efficiency operates selectively in the *TP53* mutant tumor. However, to know whether the anti-tumor enhancement occurs in every *TP53* mutant tumor needs to probe each cancer lines, which represents a daunting task and is beyond the scope of our study. Still, the treatments combination did not compromise the CDDP antitumor efficacy in *TP53* wt xenograft (HBCx-90). In this revised version, we have added new data showing effective hearing protection with local intratympanic injection of PFT-a, a route suitable for clinical practice in any types of CDDP-based cancer therapy.

See result section: page 9, line 24-25 and page 10, line 1-7 for local application of PFT- α , through an intratympanic injection;

See page 10, line 9-24 and page 11, line 1-8 for our new data obtained with *TP53* wt HBCx-90 tumor-bearing mice;

See page 12, line 14-17 for cleaved caspase 3 positive cells;

and page 13, line 6-25 and page 14, line 1-2 for autophagy.

And discussion, page 16, line 17-22; page 17, line 24-25 and page 18, line 1-18.

5) Reviewer comments

Xenograft experiments are performed using p53-mutated xenografts (Marangoni et al, 2007). In these tumors, PFT-alpha does not counteract the cisplatin toxicity and surprisingly even enhances its effect. A possible alternative explanation is that the mutant p53 protein aids the survival of the tumor after cisplatin exposure and inhibition of (mutant) p53 sensitizes the xenograft for cisplatin. The general conclusion that reversible p53-inhibition enhances antitumor efficacy and inhibits tumor recurrence (title) not justified. Experiments with different xenograft models need to be performed including p53-positive and p53-null xenografts to confirm that PFT-alpha indeed does not counteract the antitumor activity of cisplatin overall.

Authors' answer

The reviewer is correct, our title is not justified. Following the reviewer's suggestions, we generated new data using a P53-positive tumor (HBCx-90). While the anti-tumoral efficiency enhancement occurs specifically in the P53-mutated tumor, PFT-a still prevented CDDP ototoxicity in both xenografts.

These results confirmed at least in part the reviewer's hypothesis that activated mutant p53 may favour the survival of cancer cells upon CDDP intoxication. While P53-null mice exist, no triple negative patient-derived breast cancer without P53 (P53-null) is available. Thus, we could not repeat these experiments using the same human subtype xenograft model (that can be transplanted directly from the patient to the mice, *i.e.*, without *in vitro* manipulations).

However, we provide new data demonstrating the efficiency of hearing protection using local inner ear PFT-a administration, which could be suitable for CDDP-based chemotherapy in any case.

We have rewritten the results, discussion and conclusion sections and given a modified title according to these new data.

See the responses to comment 2 and 4 for the new results, discussion and new conclusion.

See also new title.

6) Reviewer comments

Figure 6A shows representative pictures of xenografts and tumor volume after the different treatments. Combination treatment completely eliminates the tumor. In the rest of the figure, the authors perform several stainings on the different tumors, including the tumor of combination treated animals. How can the authors examine tumors that are supposed to be eliminated?

Authors' answer

We apologize for this confusion. All histological analyses were performed using tumors collected only 1 week after the end of treatment (day 21). The disappearance of the majority of tumors in the combination treated group was observed around 3 weeks after the end of treatments (day 35). The procedure is now clarified in M & M.

See page 26, line 22-25 and page 27, line 1-6.

"To monitor tumor biomarkers, 3 animals from each treatment regimen and each xenograft model were randomly selected and sacrificed at day 18. For histological features of tumors and angiogenesis, 2 *TP53* wt HBCx-90-bearing mice treated with either DMSO or PFT- α alone and 3-4 animals from each of the other groups were randomly selected and sacrificed at day 21. To assess hearing function and hair cell morphologies, 5 other animals from *TP53* wt HBCx-90-bearing mice needed to be sacrificed at day 28, and from *TP53* mutant HBCx-14-bearing mice at day 35 due to tumor volume reaching around 3 cm³ in both HBCx-14- and HBCx-90-bearing mice treated with either DMSO or PFT- α alone. Five animals from the CDDP- and CDDP+PFT-a - treated *TP53* mutant HBCx-14-bearing mice were kept up to day 70 to evaluate tumor recurrence."

7) Reviewer comments

To my opinion, the figures are quite crowded and it is difficult to maintain a clear overview. The authors might replace some non-essential data to the supplemental section to create a better overview. For example x projection and 3D pictures of the IF staining.

Authors' answer

As suggested, we have replaced 3D pictures of the figure 3 to the Expanded View.

See Expanded View Figure 2C-D and its legend.

8) Reviewer comments

On page 7 is mentioned, both ATR and Chk1 mRNA are expressed... Genes are expressed and mRNA is produced and/or present.

Authors' answer

This has been corrected

-Fig 5B. The legend of p53 wt green bar is missing.

Authors' answer

Done

-Fig 5D. Why not use the same staining as in the rest of the figures (such as 1A)?

Authors' answer

In the *in vivo* part of the study, the sensory hair cell loss was evaluated using scanning electron microscopy, which is a more accurate technique not only to count the presence or absence of sensory hair cell but also to evaluate their apical pole anatomy. This technique is more complicated in practice and time consuming, so that we cannot use it for the *in vitro* hair cell quantification (such as seen in 1A).

Reviewer 2**1) Reviewer comments**

In this paper the authors nicely described new data regarding the anti-tumoral effect of p53 inhibition and its association to ototoxicity. They begin the paper showing the mechanism of ototoxicity of CDDP both in vitro and in vivo, that goes through p53 activation. Then, they studied the effect of a specific p53 inhibitor (PFT-alpha or PFT) on ototoxicity induced by CDDP treatment in a patient derived breast cancer xenograft mouse model. While, PFT protects against CDDP ototoxicity, as expected, they surprisingly found that this treatment is strikingly effective on p53 negative tumor growth. While, it has been previously demonstrated that PFT protects hair cells from cisplatin ototoxicity, the effect of PFT on tumor growth is really interesting and sheds light on new molecular mechanisms that can give rise to new treatments for cancer and ototoxicity.

Authors' answer

As with the first reviewer, the general comments of reviewer 2 are very positive. He found that our data nicely describes new data about the anti-tumoral effect of p53 inhibition and its association to ototoxicity. Our discovery of the effect of PFT on tumor growth is really interesting and sheds light on new molecular mechanisms that can give rise to new treatments for cancer and ototoxicity. We are really grateful to the reviewer for these positive comments.

2) Reviewer comments

However, there are numerous points that need to be clarified:

PFT is a p53 inhibitor and the authors showed that it could help to target p53 negative tumors. This is difficult to understand even if the authors claim: "the presence of p53 within cancer stromal cells (endothelial cells or fibroblasts) may play an important role in tumor growth". The authors have to show p53 positive cells (endothelial cells?) around the tumor. If such, how the authors can explain the absence of effect of PFT alone (figure 6D) on vascular area?

Authors' answer

P53 is generally activated under non-genotoxic and genotoxic stresses such as CDDP intoxication, this may explain at least in part the absence of effect of PFT alone. To address the point raised by the reviewer, we generated new data with immunostaining which showed more cleaved caspase 3 positive labelled stromal cells in CDDP plus PFT-a treated-TP53 mutant HBCx-14 than CDDP alone. This new data support our hypothesis that the inhibition of p53 mediated G1 cell cycle arrest and DNA damage repair with PFT-a upon CDDP intoxication probably impeding the DNA repair and hence redirecting stromal cells toward apoptosis.

See page 12, line 14-17.

“combined therapy-treated TP53 mutant tumors displayed high levels of cleaved caspase 3 positive cells in vimentin labelled stromal compartment at four days after the end of treatments (d18), when compared with CDDP alone (Fig 7C).”

3) Reviewer comments

In addition, how the authors explain that CD133 cancer stem cells - that should be p53 negative - are decreased in the presence of CDDP and PFT? That has to be explain. It really seems that there is a synergic effect of CDDP and PFT on tumor growth. This needs to be explained at molecular level, since tumors are p53 negative and since PFT has no effect alone.

Authors' answer

We apologize for this confusion. The triple negative breast cancer xenograft (HBCx-14) that we used expresses mutant p53 and contains two mutations at exon 5 of the TP53 gene. This tumor carries a p53 missense or substitution mutation, but not a p53 deletion. According to the quantitative analysis of p53 residual transactivation activity from 3000 sporadic and familial breast carcinomas included in the universal mutation database, the mean p53 residual transactivation activity in p53 mutant can reach \approx 15 to 40% of wt P53 value (http://p53.free.fr/Database/p53_DB_Breast.html).

To take the reviewer's comment into consideration, we assessed the functional integrity of the p53 pathway using western blot analysis, our new data showed that CDDP induced the stabilization of p53 and accumulation of p21, a downstream p53 target gene, also in TP53 mutant HBCx-14 tumors, but with a lower level than TP53 wt HBCx-90 tumors. These results suggest that the TP53 mutant HBCx-14 tumors retain some

p53 residual transactivation activity as previously reported in sarcoma cell lines and resected oral cavity squamous cell carcinomas from patients (Pospisilova *et al*, 2004; Perrone *et al*, 2010).

Now the detailed information concerning the p53 mutated triple negative breast cancer is added in the M & M section and these new results in the results and discussion section.

See M & M section: page 26, line 3-4, for p53 mutation information.

“The p53 mutant tumor contained two mutations at exon 5 of *TP53*: p.Y163C (Substitution - Missense) and c.488A>G (Substitution).”

And see result section: page 12, line 19-25 and page 13, line 1-2, for p53 residual transactivation activity in *TP53* mutant tumors.

“To understand how p53 inhibition potentiated the anticancer effect of CDDP in the p53 mutated cancer xenograft model, we assessed the accumulation of p53 and its downstream effector p21 in the two HBCx models. As seen in **Figure 7D, treatment of mice with CDDP resulted in a lesser stabilization of p53 and accumulation of p21 in HBCx-14 (*TP53* mutant) compared to in HBCx-90 (*TP53* wt) tumors. These results suggest that the *TP53* mutant HBCx-14 tumors retain some p53 residual transactivation activity as previously reported in sarcoma cell line and squamous cell carcinoma resected from the oral cavity of patients (Pospisilova *et al*, 2004; Perrone *et al*, 2010).”**

4) Reviewer comments

The authors based the entire study on one patient-derived xenograft (PDX). In order to prove that p53 inhibition is really effective to decrease p53 negative tumor size, the authors have to extend their study to other breast cancer PDX (that are numerous) and compare p53 positive and p53 negative tumors.

Finally, they also have to address the problem of metastasis since it has been recently shown that p53 deficiency enhances tumor growth at metastatic sites (Breast cancer res, 2016, Powell *et al*).

Authors' answer

This study was originally designed to decipher the mechanisms mediating the response of the post-mitotic cochlear cells to DNA damage upon CDDP intoxication and to counteract this ototoxicity with specific inhibitors of the DNA damage response pathway. Finally, using a pre-clinical human breast cancer xenograft model with mutated p53, we provide a proof of principle that reversible p53 suppression was feasible to prevent CDDP-induced hearing loss without compromising event-enhancing CDDP anticancer efficiency in this p53 mutated human breast cancer. Since the tumor model that we used is known to maintain the drug response of the original patient tumors (Marangoni *et al*, 2007), in this revised version, we added new data from the same subtype of human tumor (triple negative breast cancer) but with p53 status wild type (HBCx-90). No enhancement of the anti-tumor effect was observed, suggesting that the increase in the anti-tumoral efficiency operates selectively in the *P53* mutant tumor. However, to know whether the anti-tumor enhancement occurs in every *P53* mutant tumor we would need to probe each cancer line, which represents a daunting task and is beyond the scope of our study.

Nevertheless, to provide a safe and clinically achievable method of delivering combined treatments, we have added new data showing the efficiency of hearing protection through local transtympanic injections of PFT- α , which is a clinically suitable method for hearing protection in any CDDP-based cancer therapies.

See result section: page 9, line 24-25 and page 10, line 1-7 for local application of PFT- α , through an intratympanic injection;

See page 10, line 9-24 and page 11, line 1-8 for our new data obtained with *TP53* wt HBCx-90 tumor-bearing mice;

See also discussion: page 16, line 17-22 for local application of PFT- α ;

Page 17, line 6-13 for xenograft models.

5) Reviewer comments

Statistical analyses: the authors mentioned that they have done student-t-test or Wilcoxon's test. However, in the majority of the graphs there is 3 or more groups that need to be analyzed by ANOVA followed by a post test. In addition, it is not mentioned when a parametric or non-parametric test has been used. Therefore all the statistical analyses should be explained in details and corrected for data containing 3 or more groups.

Authors' answer

Done.

See M & M section: page 31, line 11-18.

“Data are mean \pm SEM. If conditions for a parametric test were met, the significance of the group differences was assessed with a one-way ANOVA; once the significance of the group differences ($p < 0.05$) was established, Tukey's **post hoc** tests were subsequently used for pairwise comparisons. If not, Kruskal–Wallis tests were used to assess the significance of differences among several groups; if the group differences were significant ($p < 0.05$), Dunn's tests were then used for **post hoc** comparisons between pairs of groups. Data analysis was performed using Matlab (The Mathworks Company) and its Statistics toolbox. * $p < 0.05$, ** $p < 0.01$, and *** $p < 0.001$.”

And see also all legends for figures

6) Reviewer comments

Figure 1

As cisplatin is a DNA intercalating drug it may be toxic for both hair cells and neurons. No images at the level of neurons are shown and should be added to see if cochlear cell death mechanism is ubiquitous or not.

Authors' answer

As CDDP preferentially poisons the sensory outer hair cells located in the basal cochlear turn (Rybak et al., 2007; Schacht et al., 2012), and as, during the cochlear explant preparation the SGN central axons are removed, thus causing a mechanical damage to SGNs, in addition to the known culture effects on neuron survival (Barclay et al., Neural Dev. 2011), we focused our study on the effect of CDDP on the basal cochlear sensory hair cells. However, in this revised version, to take the reviewer's suggestion into consideration, we did some additional experiments with CDDP-intoxicated cochlear slice. These new data are added to result and Expanded View Figure 1.

See Result page 5, line 11-13.

“In addition, we observed no significant decrease in spiral ganglion neuron density in the cochlear slices 2 days following CDDP treatment (day 5) (Fig EV1C-D).”

Page 6, line 24 and page 7, line 1-2.

“In contrast, only some spiral ganglion neurons from the CDDP intoxicated cochlear slices displayed a low level of nuclear γ H2AX and 53BP1 foci formation (Fig EV1H-I).”

Page 7, line 25 and page 8, line 1-2.

“However, in our experimental conditions, we failed to detect any activation of ATM or Chk2 in spiral ganglion neurons after CDDP intoxication.”

And also Expanded View figure 1 legend.

7) Reviewer comments

panel C: what are the different curves? there are 2 blue and 2 red (with different intensities). This is not explained in the legend and should be added.

panel D: whole cochlear extracts?

panel E: how caspase-3 level is quantified (fold activation??)

Authors' answer

We have now added all this information in the legend of Figure 1:

See Page 35, line 11-13.

“Effect over time on OHCs and IHCs treated with either culture medium alone (light blue and red lines for OHCs and IHCs, respectively), or 10 μ M CDDP (blue and red lines for OHCs and IHCs, respectively)”

See Page 35, line 16-17 and page 36, line 1-3.

“Histogram representing the change in cleaved caspase 3 expression levels over time in control and CDDP group. Actin served as a loading control.”

“data are expressed as mean \pm SEM. Statistical analyses in E: one-way ANOVA test followed by *post hoc* Tukey’s test. * p <0.05, ** p <0.01 and *** p <0.001, control group at 12 h vs. 1, 2, 3 and 5 d, and CDDP group at 12 h vs. 1, 2, 3 and 5 d.”

8) Reviewer comments

Figure 4:

panel 4D: Differential effect of KU55933 (ATM inhibitor) in OHC and IHC. it is not clear why pATM favors DSB repair in IHC while it induces apoptosis in OHC. CDDP induced DSB that induces pATM, pCHK2 and p-P53 to finally induces apoptosis. Therefore the effect of KU55933 on IHCs is really intriguing and needs to be clarified.

Authors' answer

We agree with the reviewer’s comment and clarified this point in the discussion section:

See Discussion: page 15, line 6-7 and line 13-19.

“ATM and ATR, two early sensors of DNA damage, play a crucial role in cell fate decision upon DNA damage ..., In mildly damaged cells such as the IHCs following CDDP, few γ H2AX and 53BP1 foci could be recovered through DNA damage repair mechanisms. In these cells, the inhibition of ATM would increase the number of unrepaired DNA damage foci and subsequently lead to IHC loss. By contrast, in OHCs displaying severe DNA damage as illustrated by higher levels of γ H2AX and 53BP1, p-ATM would result in the activation of the cell death pathway and OHC loss. Thus, the inhibition of ATM protected these cells from apoptosis.”

9) Reviewer comments

In general, the results regarding a differential effect on OHCs and IHCs are not clear. The authors showed in figure 1 that CDDP is not inducing any IHCs death (and clearly there is no annexin V nor tunel positive IHCs) but they then showed on figure 4H that CDDP seems to have an effect that is restore upon PFT treatment (and mentioned it in the text p8). In addition as mentioned above, statistical analysis is not clear on this figure.

Authors' answer

In Figure 1, the quantification of hair cell loss showed that CDDP exposure at 10 μ M led to a massive degeneration of OHCs, but only 8% of IHC loss in the basal turn of the organ of Corti 5 days after exposure (Figure 1C). In Figure 4h (now, EV Figure 2H), we showed the same range of CDDP-induced IHC loss which was restored upon PFT treatment. In this revised manuscript, we have added two representative images showing a small fraction of annexin V- and TUNEL- positive IHCs on the 3rd day after CDDP exposure.

See result, page 5, line 21-22.

“In contrast, only marginal IHCs showed positive annexin V or TUNEL staining (Fig EV1F-G).”

And also legend for EV figure 1E-G.

In addition, the statistical analysis that we used is now given in the legend for figure 4, see page 39, line 9-10.

“Statistical analyses in B, D and F: one-way ANOVA test followed by *post hoc* Tukey’s test. * p <0.05, ** p <0.01 and *** p <0.001.”

10) Reviewer comments

Figure 6:

Effect of PFT alone on tumor volume? (even if there is no effect on CD31 and CD133 cells).

The authors claim that CDDP +PFT treatment inhibits local tumor recurrence but they only showed one stage of recurrence (70 days) where a difference appears. They should show another stage to claim such statement.

Authors' answer

In our experiments, recurrence was evaluated over 5 stages (28, 35, 42, 56 and 70 days). Tracking the recurrence over longer periods is by nature time consuming, so we would prefer such evaluation to be the aim of another additional study.

What is the explanation for the synergic effect of CDDP and PFT on vascular area and cd133 positive cell number since PFT alone has no effect. As the authors suggest in the discussion that vascular cells and stem cells may be p53 positive it is difficult to understand the absence of effect of PFT alone.

Authors' answer

Generally, P53 is activated under non-genotoxic or genotoxic stresses such as anticancer drug intoxication (Vousden & Prives, 2009; Bursac *et al*, 2014). This may explain at least in part the absence of any effect of PFT alone (without CDDP poisoning).

11) Reviewer comments

Mat and met:

numerous sections need additional details:

- animals: why do the authors used swiss mice for in vitro experiments and not mice from the same background as p53-KO mice (129sv).

Authors' answer

The choice of Swiss mice to examine the *in vitro* signalling pathway has been dictated from previous studies (Wang *et al.*, J Neurosci. 2003; Munnamalai *et al.*, J Neurosci. 2012). Switching to *in vivo* models obviously forced us to make further investigations in a different background. We agree that using mice from the same background would be easier for the reader but on the other hand, showing that P53 inhibition is critical for hearing protection in different mouse lines make our results more robust and more confident as there are independent of the mouse background.

- Construction of p53 ko mice should be described a little (at least a good reference should be added since the authors cited a wrong ref from jackson and raymond that deals with MITF and not p53 and that is a comment on two other papers!!!)

Authors' answer

Done, we apologize for this mistake.

See M & M section: page 20, line 4-7.

“p53^{-/-} mice back-crossed into the isogenic 129sv background were a gift from the [Institute of Molecular Genetics of Montpellier](http://transgenose.cnrs-orleans.fr/taam/cda.php), but were [originally obtained from Transgenesis, Archiving and Animal Models Centre](http://transgenose.cnrs-orleans.fr/taam/cda.php) (see: <http://transgenose.cnrs-orleans.fr/taam/cda.php>).”

See also page 24, line 22 for reference, “Jacks *et al*, 1994 ”.

12) Reviewer comments

mice bearing xenograft: the authors should explained the rationale to used this specific p53-Rb-PTEN triple-neg breast cancer. It seems that the authors could have used many other HBCx tumors as described previously (breast cancer research, 2014, grinde *et al.*) or even at jackson lab where numerous tumors are commercialized.

Authors' answer

The rationale behind using triple negative patient-derived breast cancer xenograft mouse models are now given in the discussion.

See page 17, line 6-10.

“The tumor xenograft models that we used in the present study were obtained from directly transplanting patient-derived tumor fragment into mice. In contrast to cell line-derived xenografts, these human-to-mice tumor xenografts maintain the cell differentiation, morphology, and drug response properties of the original patient tumors (Marangoni *et al*, 2007).”

13) Reviewer comments

- How many cells have been graft and where? This should be clarified.

Authors' answer

Done,
See M & M section: page 26, line 4-9.

“The generation of these xenografts has been published previously (Marangoni *et al*, 2007). Briefly, tumor specimens were obtained from consenting patients during surgical resection. The tumor samples were established as xenografts by subcutaneous implantation of a tumor fragment into the interscapular fat pad of Swiss nude mice. They were subsequently transplanted from mouse to mouse with a volume of approximately 15mm³.”

14) Reviewer comments

- What is the brand of the apparatus allowing auditory function recording?
 the authors mentioned: "The ABR thresholds were defined as the minimum sound intensity necessary to elicit a clearly distinguishable response (> 0.2μV)". They should clarify what does this mean? For which peak? is it a blind analysis?

Authors' answer

Done,

See M & M section: page 27, line 25 and page 28, line 1-11.

“Auditory brainstem responses were recorded from three subcutaneous needle electrodes placed on the vertex (active), on the pinna of the tested ear, and in the neck muscles (ground) of the mice. The acoustical stimuli were generated by a NI PXI-4461 signal generator (National Instruments) consisting of 10 ms tone-bursts with a 1 ms rise and fall time delivered at a rate of 10/s. Sound was delivered by a JBL 075 loudspeaker (James B. Lansing Sound) in a calibrated free-field condition, positioned at 10 cm from the tested ear. Cochlear amplification (20,000) was achieved via a Grass P511 differential amplifier, averaged 1000 times (Dell Dimensions). Intensity-amplitude functions of the ABRs were obtained at each frequency tested (4, 6.3, 8, 10, 12.5, 16, 20, 25, and 32 kHz) by varying the intensity of the tone bursts from 0 to 100 dB SPL, in 5dB incremental steps. The ABR thresholds were defined as the minimum sound intensity necessary to elicit well-defined and reproducible wave II. Recordings and analysis were performed blindly.”

- Did the hair cell counts were done in the same way for p53 and PDX mice? not clear in the text

Authors' answer

Clarification is now given in M & M section.

See page 28, line 16-19 for hair cell counting in p53wt and p53-/- mice.

And page 28, line 19-24 for HBCx models.

“In all Swiss nude mice bearing xenografts or not, a massive loss of OHCs was observed in the extreme basal cochlear region, independent of CDDP treatment. Therefore, in these xenograft mice, to evaluate specific hair cell loss induced by CDDP, the counting was restricted to three different 300 µm long segments of the organ of Corti, centered at 1.1, 2.6, 3.5 mm from the cochlear apical region.”

15) Reviewer comments

- catalog number for drugs that have been used should be added

Authors' answer

Done.

See M & M section.

- cultures : how the explants are maintained in 6 well plates? In suspension or in adherent conditions?

Authors' answer

We have added this information in M & M section, page 21, line 17-18.

“The whole cochleae were kept in suspension, and the organ of Corti explants and cochlear slices in adherent conditions”

- Detailed info for antibodies should be added (catalogue number, exact name of the society: for example, molecular probes does not exists anymore, it is fisher that distributes the corresponding products).

Authors' answer
Done, thanks for reviewer's comments and suggestions!

2nd Editorial Decision

09 September 2016

Thank you for the submission of your revised manuscript to EMBO Molecular Medicine. We have now received the enclosed reports from the referees that were asked to re-assess it. As you will see the reviewers are now fully supportive and I am pleased to inform you that we will be able to accept your manuscript pending final editorial amendments.

Please submit your revised manuscript within two weeks. I look forward to seeing a revised form of your manuscript as soon as possible.

***** Reviewer's comments *****

Referee #1 (Comments on Novelty/Model System):

Every model has its limitations but the experiments in the revised manuscript involve highly relevant models. Evidently future research should determine to which extent the findings made here are applicable in the case of patients.

Referee #1 (Remarks):

In the revised version of the manuscript, the authors have addressed in an adequate manner all questions and concerns raised. New data have been added and overstatements have been toned down or otherwise adapted and major conclusions on mutant and wt P53 have been modified on the basis of new experimental results. This has substantially improved the manuscript. In my view the revised paper now meets the high standards for publication in EMBO molecular medicine.

Referee #2 (Remarks):

The authors substantially improved the paper and answer the majority of the questions. However, the originality in this paper is NOT the fact that PFT protects against cisplatin toxicity, which has ever been shown even on cultured organs of Corti, but the fact that PFT does not interfere with cisplatin anti-mitotic effect and much more that cisplatin antitumoral effect is increased by PFT. This is in line with a recent paper showing that an analog of PFTalpha (PFTmu) acts synergistically with platinum derivatives on cancer cells (Molecules, 2016, McKeon et al.). This should have been studied a little further in terms of mechanism as stated by the attractive title of the paper in its first version. Now the title and the message is a little less original.

Corresponding Author Name: Jing Wang
Journal Submitted to: EMBO Molecular Medicine
Manuscript Number: EMM-2016-06230-V2